# Continuous Space-Time Video Super-Resolution with 3D Fourier Fields

**Alexander Becker**    **Julius Erbach**    **Dominik Narnhofer**    **Konrad Schindler**

ETH Zurich

## Abstract

We introduce a novel formulation for continuous space-time video super-resolution. Instead of decoupling the representation of a video sequence into separate spatial and temporal components and relying on brittle, explicit frame warping for motion compensation, we encode video as a continuous, spatio-temporally coherent 3D Video Fourier Field (VFF). That representation offers three key advantages: (1) it enables cheap, flexible sampling at arbitrary locations in space and time; (2) it is able to simultaneously capture fine spatial detail and smooth temporal dynamics; and (3) it offers the possibility to include an analytical, Gaussian point spread function in the sampling to ensure aliasing-free reconstruction at arbitrary scale. The coefficients of the proposed, Fourier-like sinusoidal basis are predicted with a neural encoder with a large spatio-temporal receptive field, conditioned on the low-resolution input video. Through extensive experiments, we show that our joint modeling substantially improves both spatial and temporal super-resolution and sets a new state of the art for multiple benchmarks: across a wide range of upscaling factors, it delivers sharper and temporally more consistent reconstructions than existing baselines, while being computationally more efficient. Project page: `https://v3vsr.github.io`.

## 1 Introduction

Video super-resolution (VSR) seeks to improve the perceptual quality by reconstructing high-resolution (HR) videos from low-resolution (LR) inputs. VSR is an elementary capability for video editing and analysis, because the spatial resolution and frame rate of video recordings are limited by hardware and power constraints. For instance, mobile devices and action cameras typically lack a continuous optical zoom and rely on VSR for digital enlargement.

To be practical, a super-resolution scheme should not be tied to a specific magnification, but support the reconstruction of videos with *arbitrary upsampling factors*, in both space and time. Several recent works have addressed this by representing video as a continuous function – often an implicit neural representation (INR).

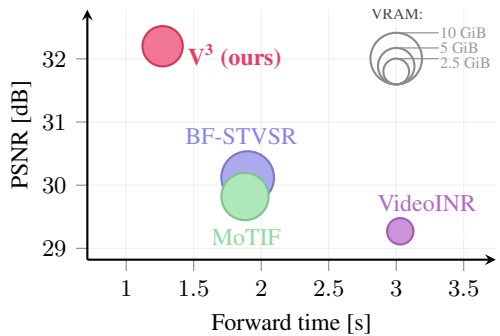

Figure 1: **Performance vs. computation time.** $V^3$ outperforms other VSR models by about 2 dB PSNR, while being significantly faster. PSNR is measured on the Adobe240 test set, for $\times 4$ spatial and $\times 8$ temporal SR. For compute see Sec. 4.5.

Once that function has been inferred from the available (low-resolution) observations, it can be sampled at any desired locations, respectively raster spacing. Existing implementations of that principle, however, share an important limitation: they separate spatial and temporal modeling, i.e., each frame pair is spatially represented by a 2D function (typically a 2D INR), and the motion in between frames is represented by another function (typically an optical flow field). This factorization is conceptually unsatisfactory, since it risks losing spatio-temporal correlations. Perhaps more importantly, it is also practically undesirable: to exchange information between different frames one

must rely on explicit warping, making super-resolution vulnerable to errors of the underlying flow estimation – which tend to be worst in critical regions, e.g., near object boundaries. To make matters worse, optical flow is estimated from two consecutive images. "Temporal" modeling often does not go beyond pairs of adjacent frames, because chaining or fusing flow vectors over longer, more meaningful temporal contexts is hard due to error build-up, over-smoothing and (dis-)occlusions.

An important capability for an arbitrary-scale super-resolution method is correct *anti-aliasing*: at the time when the representation is learned, it is not known at which scales it will later be sampled. Consequently, the representation must contain high-frequency detail up to the highest possible upsampling factor, which means that, when super-resolving by lower factors, those details will lie beyond the Nyquist limit. To avoid artifacts, the sampling mechanism must be designed in a way that suppresses unrepresentable frequencies. With INRs, this is complicated: their representation resides in an abstract latent feature space that is hard to manipulate. Sampling them with an integrative observation model (a "point spread function") requires workarounds that add complexity and computational cost.

To tackle these challenges, we resort to a drastically simpler representation: a combination of 3D sine waves in $(x, y, t)$. I.e., we sidestep the above-mentioned problems by *jointly* embedding the space and time dimensions in a single, unified representation. Our VFF representation – to our knowledge, the first such model for video super-resolution – is conceptually simple and interpretable, and at the same time a natural choice if one aims to decode the video at arbitrary spatial and temporal scales: Like previous methods, VFF can be queried at any desired, continuous spatio-temporal coordinate – but at much lower computational cost. However, VFF avoids the potentially error-prone, warping-based decoupling of spatial and temporal modeling, and is, by construction, amenable to longer-term temporal modeling over multiple frames. What is more, translational motions correspond to phase shifts in VFF, making it easy to capture (and learn) a dominant component of video motion. Moreover, the functional form of VFF permits closed-form sampling with Gaussian PSFs, important for correct anti-aliasing when moving across scales, especially in out-of-distribution settings.

Intuitively, VFF can be thought of as a continuous $(x, y, t)$-cuboid that can be queried at arbitrary spatio-temporal locations to obtain a video with the desired resolution and frame rate. To map a (low resolution, low frame rate) input video to that cuboid, we predict the corresponding basis coefficients, using a neural video backbone. Inference of the super-resolved video amounts to sampling the representation at the corresponding spatio-temporal grid points.

The resulting, end-to-end trainable continuous space-time video super-resolution (C-STVSR) method, which we name $V^3$, outperforms existing baselines by a substantial margin across different tasks and datasets, while being computationally more efficient (Fig. 1). In summary, our **contributions** are:

- VFF, a radically simple, yet highly effective continuous-domain video representation, consisting of a single trigonometric expansion of the joint $(x, y, t)$ space.

- $V^3$, an end-to-end framework to predict the parameters of VFF directly from a low-quality input video, using a backbone encoder with large spatio-temporal receptive field.

- An extensive experimental evaluation across different super-resolution tasks, in which $V^3$ outperforms prior C-STVSR approaches by up to $\approx 2\,\text{dB}$ in PSNR, while at the same time reducing runtime and memory footprint.

## 2 RELATED WORK

*Space-time video super-resolution* (STVSR) aims to enhance both spatial resolution and frame rate in a unified framework. A straightforward baseline for STVSR is a two-stage pipeline: first apply a video frame interpolation method such as SuperSloMo (Jiang et al., 2018) or DAIN (Bao et al., 2019), then follow up with a single-image or video SR model such as RCAN (Zhang et al., 2018) or EDVR (Wang et al., 2019). While intuitive, this separation ignores correlations across space and time and often introduces temporal flicker. To avoid it, one-stage STVSR methods such as Zooming Slow-Mo (Xiang et al., 2020) and TMNet (Xu et al., 2021) perform spatial and temporal upsampling jointly. These methods generally assume *fixed, integer upscaling factors*, limiting their usability.

To go beyond fixed scales requires *arbitrary-scale super-resolution*, which has predominantly been studied for *single images*. Early work relied on meta-learning (Hu et al., 2019) or implicit neural representations (INRs, e.g. Chen et al., 2021) to enable sampling at arbitrary resolutions without retraining. Lee & Jin (2022) employ a Fourier basis within such an INR framework, while Becker et al. (2025) introduce a learned, non-orthogonal sinusoidal basis for fast and theoretically guaranteed anti-aliasing. Naive frame-by-frame application of image super-resolution to videos ignores temporal dependencies and leads to flickering. Hence, dedicated methods have been proposed for *arbitrary-scale video SR* (AVSR), which incorporate temporal guidance features (Li et al., 2024; Shang et al., 2024). These methods, however, do not support temporal upsampling.

So-called *continuous STVSR* (C-STVSR) represents both space and time as continuous domains and is therefore able to upsample in space and in time. It includes AVSR as well as fixed-scale STVSR (and also single-image SR) as special cases and, arguably, constitutes the most general and practical formulation. C-STVSR was pioneered by VideoINR (Chen et al., 2022), where videos are parametrized as two separate, decoupled INRs in image space and time. The temporal INR acts as an optical flow predictor, estimating the motion field from the intermediate frame at time $t$ to the keyframe as a function of $t$, followed by backward warping to reconstruct intermediate features. However, modeling continuous backward flow fields is challenging: a location's content – and thus its motion – varies with time. Even under linear motion, the backward flow field changes structurally over time, producing discontinuities at motion boundaries that are challenging to learn. In contrast, forward flow (key-to-intermediate) tracks each pixel's trajectory through time as a continuous curve. For linear motion, the displacement vector field preserves its structure and simply scales with t, making it easier to model. Building on this idea, MoTIF (Chen et al., 2023b) learns forward motion trajectories and applies softmax splatting (Niklaus & Liu, 2020) for forward warping, leaving occlusion handling and conflicting motions to the decoder. BF-STVSR (Kim et al., 2025) incorporates a Fourier basis in the latent space and B-splines to parametrize the motion field, but still depends on explicit warping to map the two keyframes to the intermediate. Furthermore, none of the mentioned methods has a principled mechanism for anti-aliasing. Instead, they count on implicitly learning the necessary, adaptive high-pass filtering from data, which makes the learning problem significantly more complicated and comes without any guarantees.

In summary, the existing literature shows a growing interest in C-STVSR, but exposes a clear conceptual gap: no existing method provides a unified, spatio-temporally consistent representation that would (i) keep the system design simple; (ii) circumvent explicit, error-prone warping; (iii) allow for multi-frame motion context; and (iv) include an efficient mechanism for the elementary anti-aliasing operation.

## 3 METHOD

To describe our formulation for continuous space-time video super-resolution we first define the problem setup, then introduce our VFF video representation and its conditional parameterization, and finally show how to embed it in an end-to-end learning framework.

### 3.1 PROBLEM STATEMENT

Let $\mathbf{V}^{lr} \in \mathbb{R}^{T \times H \times W \times 3}$ denote a low-resolution RGB video. The objective of C-STVSR is to recover a continuous spatio-temporal signal,

$$\hat{V}(x, y, t) : \mathbb{R}^2 \times [0, T] \to \mathbb{R}^3 . \tag{1}$$

The observed video satisfies $\boldsymbol{V}^{lr} = \boldsymbol{\mathcal{D}}(\hat{V})$, where $\boldsymbol{\mathcal{D}}$ describes the degradation – in the case of C-STVSR the discrete sampling of the signal in space and time. Once the signal $\hat{V}$ has been recovered, it can be sampled at an arbitrary, smaller grid spacing ($s\times$ smaller in space, $r\times$ smaller in time) to obtain a high-resolution, high-frame rate video $\boldsymbol{V}^{hr} \in \mathbb{R}^{rT \times sH \times sW \times 3}$. To reconstruct $\hat{V}$ one must invert $\boldsymbol{\mathcal{D}}$, which is highly ill-posed and only possible if one has access to a strong prior for $\hat{V}$. In modern super-resolution schemes, including $V^3$, that prior – i.e., generic expectations about the spatio-temporal structures and patterns in natural videos – takes the form of a neural network and is learned from data.

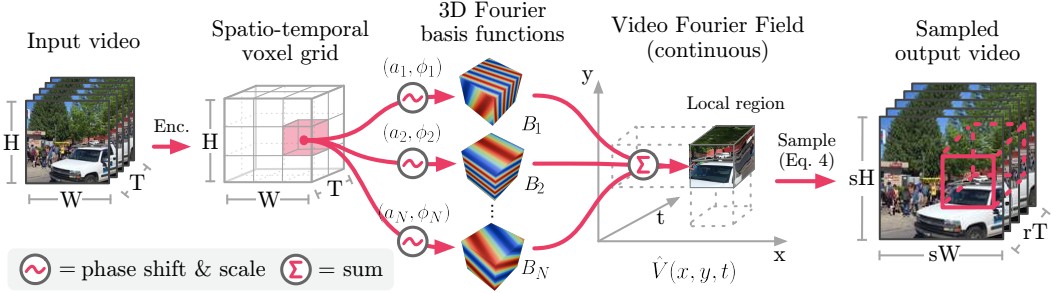

Figure 2: **Overview of V³.** A backbone encoder predicts a voxel grid of local phase shifts and weighting coefficients for a set of 3D Fourier basis functions. Their sum describes, within a local interval, the continuous function $\hat{V}(x, y, t)$ that we call the *Video Fourier Field*. The function can be sampled at different spatio-temporal resolutions (Eq. 4) to obtain an output video.

Below, we introduce a compact, but expressive parametrization of $\hat{V}$ (Sec. 3.2), followed by a practical algorithm to estimate its free parameters from training data (Sec. 3.3).

## 3.2 VIDEOS AS 3D FOURIER SERIES

The core of our representation is a collection of sinusoidal 3D basis functions $\{B_i\}_{i=1}^N$ in $(x, y, t)$-space, parameterized by frequencies $\boldsymbol{\omega}_i \in \mathbb{R}^3$ and phase shifts $\phi_i \in \mathbb{R}$:

$$B_i : \mathbb{R}^3 \to \mathbb{R} \quad , \quad (x, y, t) \mapsto a_i \cdot \sin\left(\boldsymbol{\omega}_i \cdot (x, y, t) + \phi_i\right) . \tag{2}$$

With their help, we parametrize the continuous video signal as a finite trigonometric expansion,

$$\hat{V}(x, y, t) = \sum_{i=1}^N B_i(x, y, t, a_i, \phi_i, \boldsymbol{\omega}_i), \tag{3}$$

which we refer to as a *Video Fourier Field* (VFF). We call our representation a Fourier Field because it expresses the video as a finite sum of sinusoidal functions, inspired by the classical Fourier transform. Unlike a true Fourier series, consisting of an infinite set of orthogonal functions with integer frequencies to guarantee completeness, our formulation employs a finite set of sinusoids with continuous frequencies and phase shifts. This relaxation sacrifices strict orthogonality but retains the key advantage of Fourier features: a continuous, band-limited representation that can be queried at arbitrary spatio-temporal resolutions.

Note that $\hat{V}$ is defined on a continuous domain, and can thus be sampled at arbitrary rates both spatially and temporally, as needed for C-STVSR. In practice, to keep the number of basis functions small, we split the $(x, y, t)$-space into local, axis-aligned regions ("voxels") and fit individual VFFs to them. In this way, the coefficients $(a_i, \phi_i)$ can be adjusted to the local video content, but the overall representations still covers the whole, continuous domain.

One advantage of the VFF is that it can easily be queried with a linear point spread function (PSF) to achieve correct anti-aliasing for any desired scale factor. Following Fourier theory,

$$\hat{V}_\sigma(x, y, t) = \sum_{i=1}^N B_i(x, y, t) \cdot \xi(\boldsymbol{\omega}_i, \sigma), \tag{4}$$

exactly describes $\hat{V}$ under a Gaussian PSF with variance $\sigma$. Aliasing-free sampling thus amounts to simply rescaling the individual basis functions by the frequency-dependent factor $\xi(\boldsymbol{\omega}_i, \sigma) = \exp(-||\boldsymbol{\omega}_i||^2/8\pi^2\sigma^2)$, where $\sigma$ is inversely proportional to the effective sampling rate, as dictated by the Nyquist limit (see, *e.g.*, Oppenheim et al., 1997). Note that Eq. 4 can be implemented as a matrix multiplication followed by element-wise addition (for the shift) and scaling, which is much more efficient than explicit filtering or oversampling. Compared to existing C-STVSR methods (Chen et al., 2022; 2023b; Kim et al., 2025), where a scale-appropriate PSF has to be learned from data as

part of the neural model, hard-wiring the signal-theoretically correct PSF is not only more parameter-efficient, but also ensures good generalization, unaffected by training biases, cf. Becker et al. (2025). Note also, $\sigma$ can be set to $\approx 0$ for individual dimensions, for instance to express a Gaussian blur in space but point sampling in time, or to a small constant value representing the narrow temporal PSF induced by finite exposure times in many consumer cameras. Contrary to other C-STVSR methods (Chen et al., 2022; 2023b; Kim et al., 2025), our trigonometric expansion over $(x, y, t)$ can be sampled at arbitrary spatial and temporal resolutions with a single function call, without explicit frame warping. The formulation can naturally encode translational motion as simple phase shifts in the frequency domain (Kuglin, 1975), and the sinusoidal basis compactly captures frequency patterns across scales.

### 3.3 CONDITIONAL FOURIER PARAMETERIZATION

Building on the VFF formulation, we now have to infer the parameters of the representation $\hat{V}$ for an input video, so that higher-resolution output videos can be sampled from it. We employ a domain-specific neural video encoder, $\boldsymbol{E}$, that aggregates semantic features of every input voxel over a large spatio-temporal receptive field. Unlike approaches that are restricted to pairwise optical flow, our model by design leverages a substantially larger temporal context and can reason jointly over multiple frames. Consequently, the representation can capture long-range dependencies, handle occlusions and disocclusions more robustly, and capture non-linear and periodic motion patterns that a simple frame-to-frame interpolation cannot adequately handle (see Fig. 5).

As mentioned above, in practice we use a voxel grid of local basis expansions, in line with other SR methods that do the same with local INRs (Chen et al., 2021; 2023b; Kim et al., 2025). We emphasize that each grid cell still contains only a single, unified and continuous function $(x, y, t)$. Global consistency across cell boundaries is maintained through the backbone encoder, which aggregates information over large spatio-temporal receptive fields to predict the local field parameters. Concretely, after obtaining a grid of semantic features $\boldsymbol{E}(x) \in \mathbb{R}^{T \times H \times W \times F}$ from the encoder, matching the size of the input video, we apply a small convolutional network to map those features to a 3D grid of VFF parameters, $\{(\boldsymbol{a_j}, \boldsymbol{\phi_j})\}_{\boldsymbol{j}}$, where $\boldsymbol{a_j} = (a_1, \ldots, a_N)_{\boldsymbol{j}}$ and $\boldsymbol{\phi_j} = (\phi_1, \ldots, \phi_N)_{\boldsymbol{j}}$ define the VFF at each voxel index $\boldsymbol{j} \in \mathbb{N}^3$. We find that it is not necessary to locally vary the frequencies $\{\boldsymbol{\omega}\}_{i=1}^N$ between different grid cells. In other words, the base frequencies are learned once during training. At inference time they are kept fixed for all input videos and all cells, and only their amplitudes and phases are modulated to fit the input. Beyond saving compute, sharing a common frequency basis turns out to slightly improve stability and coherence of the reconstructed video.

The complete super-resolution system (Fig. 2) consists of the backbone encoder, the VFF basis, and the PSF-aware sampler (Eq. 4). It is differentiable, and trained end-to-end.

### 3.4 IMPLEMENTATION AND TRAINING

$V^3$ is implemented in JAX (Bradbury et al., 2018). We use $N = 512$ basis functions. As backbone encoder we employ RVRT (Liang et al., 2022), with the embedding dimension set to 90 and 12 attention heads. During training, the spatial upsampling factor is sampled randomly from the interval $\mathcal{U}(1.2, 4)$, and input data is generated with bicubic downsampling in space and subsampling of (high-speed) video frames in time, as in prior work. Training patches have $80 \times 80$ pixels and 14 frames. We use standard data augmentations (random flipping, rotation, and resizing) and train for $2.5 \times 10^6$ iterations with an $L_1$ reconstruction loss using the AdamW (Loshchilov & Hutter, 2017) optimizer (lr= $10^{-4}$, $\beta_1 = 0.9$, $\beta_2 = 0.999$, $\epsilon = 10^{-8}$) and a Cosine Annealing scheduler (Loshchilov & Hutter, 2016). We furthermore employ gradient clipping at a global $L_2$ norm of 1. As in prior work, all parameters are trained from scratch, except for the flow component in RVRT (RAFT, Teed & Deng, 2020), which we finetune only for the last $3 \times 10^5$ iterations. We use a batch size of 16 and train on $16\times$ Nvidia GH200 chips. Inference only requires a single consumer GPU (RTX 3090 Ti).

## 4 EXPERIMENTS

We evaluate $V^3$ on several benchmarks. Besides C-STVSR performance, we also test the special cases of spatial-only and temporal-only SR, as well as temporal consistency. Following the

Table 1: Quantitative results for spatio-temporal super resolution (PSNR↑ / SSIM↑). The spatial scaling factor is set to ×4, the temporal to ×8 (Adobe and GoPro) and ×2 (Vid4). Columns ending in *Center* are evaluated on key- and center frames only, and *Average* denotes all output frames. We show a two-stage method for comparison (†).

| Method | Vid4 | GoPro *Center* | GoPro *Average* | Adobe *Center* | Adobe *Average* | # Par. |
|---|---|---|---|---|---|---|
| DAIN→EDVR † | 23.48 / 0.654 | 28.58 / 0.842 | 26.64 / 0.798 | 27.45 / 0.809 | 25.64 / 0.759 | 44.7 M |
| ZoomingSloMo | 25.72 / 0.772 | 30.69 / 0.885 | — | 30.26 / 0.882 | — | 11.1 M |
| TMNet | 25.96 / 0.780 | 30.14 / 0.870 | 28.83 / 0.851 | 29.41 / 0.852 | 28.30 / 0.835 | 12.3 M |
| VideoINR | 25.61 / 0.771 | 30.26 / 0.879 | 29.41 / 0.867 | 29.92 / 0.875 | 29.27 / 0.865 | 11.3 M |
| MoTIF | 25.79 / 0.775 | 31.04 / 0.888 | 30.04 / 0.877 | 30.63 / 0.884 | 29.82 / 0.875 | 12.6 M |
| BF-STVSR | 25.85 / 0.777 | 31.17 / 0.890 | 30.22 / 0.880 | 30.83 / 0.888 | 30.12 / 0.880 | 13.5 M |
| $V^3$ | 26.76 / 0.818 | 32.96 / 0.923 | 32.26 / 0.919 | 32.91 / 0.922 | 32.29 / 0.917 | 13.7 M |
| $V^3$-Large | **26.82 / 0.821** | **33.09 / 0.925** | **32.36 / 0.921** | **33.08 / 0.924** | **32.45 / 0.919** | 20.6 M |

literature (Chen et al., 2022; 2023b; Kim et al., 2025), we train on the 240 fps Adobe240 (Su et al., 2017) dataset, which consists of 133 videos at resolution 1280×720 pixels. Input videos are obtained by spatial downsampling with random scaling factors and fixed temporal subsampling by a factor of ×8 to obtain input videos with 30 fps; all ground truth frames, randomly sampled in space and time serve as ground truth for supervision.

## 4.1 C-STVSR PERFORMANCE

We begin by evaluating $V^3$ on the full, spatio-temporally continuous C-STVSR task. Table 1 shows quantitative results for various methods on the test sets of Vid4 (Liu & Sun, 2011) (spatial ×4, temporal ×2) GoPro (Nah et al., 2017) and Adobe240 (both spatial ×4, temporal ×8). We follow the evaluation protocol of previous works and compute PSNR and SSIM in the luminance channel, one for the input and center frames ("*Center*"), and once for all output frames ("*Average*"). We find that $V^3$ sets a new state of the art on all three datasets and outperforms the baselines by a substantial margin – in most cases $> 1.5$ dB in PSNR. It appears that our unified Fourier representation is able to recover more spatial detail and reconstruct motion more coherently than methods that factor the representation into a spatial and temporal component.

For a fair comparison, we have matched the capacity of $V^3$ to recent competitors, yet ≈14M parameters is a small model by modern standards. To check the scalability of our model, we therefore also train a version with a higher parameter count ($V^3$-Large, 20.6M parameters), obtained by increase the size of the attention embedding in the backbone encoder to 144. This further boosts performance, albeit with diminishing returns, and often outperforms prior art by a full 2 dB in PSNR. The experiment suggests that current models have not yet reached a saturation point, and super-resolution quality could still be increased by further scaling up the model.

Figure 3 shows a visual comparison (on non-keyframes) for a sample from the Adobe240 dataset. $V^3$ faithfully reconstructs high-frequency content over time. In the example, it is the only method that recovers both legible text and the characteristic "accordion" structure of the articulated bus joint.

## 4.2 DECOUPLING SPATIAL AND TEMPORAL SR

Our core contribution is a scale-agnostic, spatio-temporally unified video representation. $V^3$ is not in any way specialized to purely spatial arbitrary-scale video super-resolution (AVSR, no temporal upsampling) or to video frame interpolation (VFI, no spatial upsampling). Therefore, these "edge cases" are of particular interest, and they can be easily realised by simply setting the temporal, respectively spatial upsampling factor to ×1.

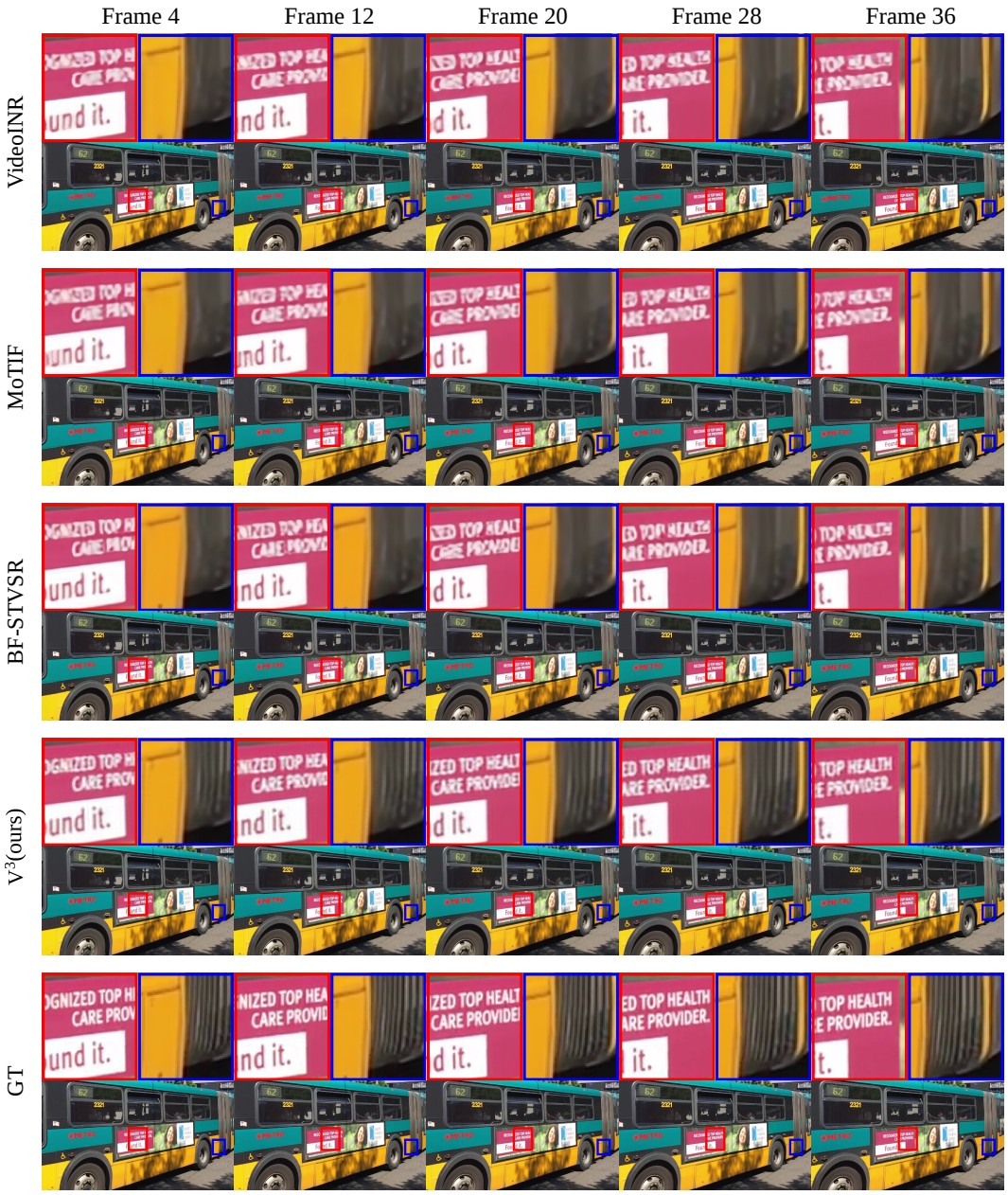

Figure 3: Qualitative comparison of C-STVSR methods ($\times 4$ spatial, $\times 8$ temporal). $V^3$ recovers legible text as well as thin stripe patterns.

### 4.2.1 ARBITRARY-SCALE VIDEO SR

To measure AVSR performance, we follow the experimental settings of the dedicated AVSR literature (Li et al., 2024; Shang et al., 2024) and evaluate on the REDS (Nah et al., 2019) validation set, consisting of 240 videos with $1280 \times 720$ pixels and 100 frames, captured with a GoPro camera at 24 fps. Table 2 compares $V^3$ to other C-STVSR methods (all trained on Adobe240) in this setting. It has been reported (Shang et al., 2024) that, when operating at $\times 1$ temporal upsampling, C-STVSR methods barely outperform arbitrary-scale *single-image* super-resolution (AISR). We therefore include three methods representative of three generations of AISR: CNN-based (LTE, Lee & Jin, 2022), modern transformer-based (CLIT, Chen et al., 2023a), and the – at the time of writing

Table 2: Spatial video super resolution (AVSR) on the REDS validation set (PSNR↑ / SSIM↑). Methods marked with † indicate arbitrary-scale *image* SR applied frame-by-frame for comparison.

| Method | $\times 2$ | $\times 3$ | $\times 4$ | $\times 6$ | $\times 8$ | # Par. |
|---|---|---|---|---|---|---|
| Bicubic | 31.51 / 0.911 | 26.82 / 0.788 | 24.92 / 0.713 | 22.89 / 0.622 | 21.69 / 0.574 | — |
| RDN-LTE † | 34.73 / 0.943 | 30.73 / 0.866 | 28.75 / 0.804 | 26.56 / 0.718 | 25.24 / 0.669 | 22.5 M |
| RDN-CLIT † | 34.63 / 0.942 | 30.63 / 0.865 | 28.63 / 0.801 | 26.43 / 0.714 | 25.14 / 0.661 | 37.7 M |
| RDN-HIIF † | 34.57 / 0.942 | 30.59 / 0.864 | 28.60 / 0.799 | 26.44 / 0.712 | 25.15 / 0.659 | 23.2 M |
| VideoINR | 31.59 / 0.900 | 30.04 / 0.852 | 28.13 / 0.791 | 25.27 / 0.687 | 23.46 / 0.619 | 11.3 M |
| MoTIF | 31.03 / 0.898 | 30.44 / 0.862 | 28.77 / 0.807 | 25.63 / 0.698 | 25.12 / 0.664 | 12.6 M |
| BF-STVSR | 34.72 / 0.946 | 31.17 / 0.881 | 29.11 / 0.820 | 26.78 / 0.728 | 25.40 / 0.668 | 13.5 M |
| $V^3$ | 36.53 / 0.963 | 32.31 / 0.908 | 29.92 / 0.849 | 27.39 / 0.754 | 25.96 / 0.690 | 13.7 M |
| $V^3$-Large | **36.70 / 0.964** | **32.53 / 0.911** | **30.13 / 0.853** | **27.54 / 0.760** | **26.10 / 0.696** | 20.6 M |

Table 3: Decoupling of spatial-only (S$\times$4, T$\times$1) and temporal-only (S$\times$1, T$\times$8) VSR on Adobe240. Input sequences have 30 fps.

| Setting | VideoINR | MoTIF | BF-STVSR | $V^3$ |
|---|---|---|---|---|
| S$\times$4, T$\times$1 | 31.84 / 0.904 | 32.95 / 0.916 | 33.03 / 0.917 | **34.25 / 0.938** |
| S$\times$1, T$\times$8 | 24.45 / 0.712 | 28.09 / 0.843 | 29.37 / 0.867 | **33.43 / 0.936** |

– strongest method based on hierarchical positional encoding, HIIF (Jiang et al., 2025). Indeed, $V^3$ appears to be the first C-STVSR method to substantially surpass per-frame image AISR across all scaling factors within and outside the training distribution. We attribute the gain to our model's better ability to transfer information between frames. The unified spatio-temporal basis gives $V^3$ access to a larger temporal context window and lets it exploit redundancy between frames for spatial super-resolution, rather than merely avoid flickering.

For completeness, Tab. 3 (top) shows spatial-only and temporal-only upsampling for the strictly in-distribution Adobe240 test set. Other C-STVSR methods fare better in this setting, apparently they are quite sensitive to the (small) domain shift to REDS. Still, $V^3$ works best by a healthy margin.

### 4.2.2 VIDEO FRAME INTERPOLATION

We next evaluate our approach for video frame interpolation (VFI), the complementary special case where the spatial scaling is fixed to $\times$1. Table 3 (bottom) compares the pure frame interpolation capabilities of C-STVSR methods on Adobe240. For this analysis, we evaluate on the centerframes, for $\times$8 temporal super resolution.

$V^3$ once more brings a substantial performance gain, demonstrating the temporal modeling power of the underlying VFF. The improvement is also evident in the qualitative results in Fig. 4: Warping-based competitors suffer from artifacts caused by inaccurate optical flow that leads to misaligned features. Specifically, the baselines often fail to handle abrupt motion boundaries with occlusions (handrail, marked with red box) and have a tendency to incorrectly combine the two nearest input frames, producing duplicate textures (text, marked with blue box).

In contrast, VFF obviates feature warping and thereby achieves more coherent and visibly sharper frame interpolation, supporting our claim that a native motion representation in $(x, y, t)$-space is more robust than one that depends on an external optical flow estimator.

### 4.3 TEMPORAL CONSISTENCY

The larger temporal context of VFF allows us to capture non-linear dynamics. We also hypothesize that representing a video in a 3D frequency space simplifies motion modeling (Kuglin, 1975) and avoids error-prone warping. All these factors should aid temporal consistency. To illustrate this, Fig. 5 depicts the temporal profile of a vertical image column as predicted by several video super-resolution methods.

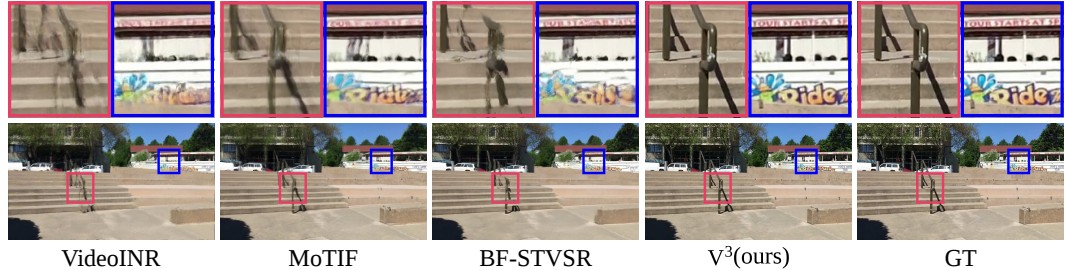

VideoINR      MoTIF      BF-STVSR      $V^3$(ours)      GT

Figure 4: Frame interpolation ($\times 8$, center frames). $V^3$ faithfully recovers high frequency content.

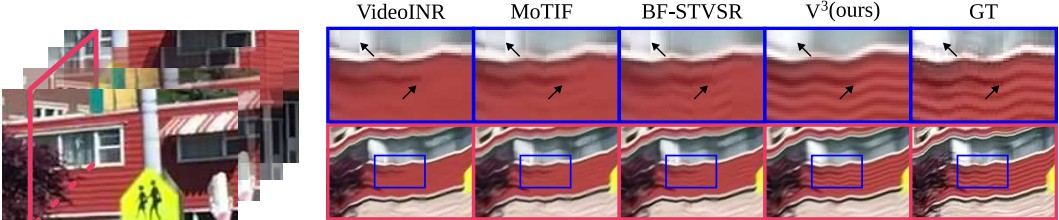

Figure 5: Temporal consistency. The red rectangle corresponds to a vertical image column over time. $V^3$ faithfully reconstructs complex, non-linear image motion and reduces block artifacts.

While $V^3$ propagates structures smoothly across time, preserving both curvature and fine details, the baselines exhibit a bias towards linear, non-periodic motion (center, wave pattern caused by the red facade), as well as blocky temporal discontinuities (top left, white window frame).

Table 4 quantitatively evaluates temporal consistency. Following Chu et al. (2018); Cao et al. (2021); Kim et al. (2025) we compute *tOF* to measure per-pixel differences in the optical flow (*i.e.*, end point error) between adjacent ground truth frames and the corresponding target frames. Intuitively, tOF quantifies how much the motion trajectories in the super-resolved video deviate from the ground truth motion, so large values imply temporal inconsistencies such as flicker or unnatural motion. The numbers confirm the superior temporal consistency of $V^3$.

### 4.4 ANALYSIS OF LEARNED BASIS FUNCTIONS

Figure 6 analyzes the spatiotemporal behavior of basis functions $B_i$ learned by $V^3$. The figure shows that $V^3$ is able to pick up structure in the data, with a higher concentration of components along the coordinate axes of the polar plot (*i.e.*, horizontal and vertical wave patterns representing axis-aligned structures). In terms of the marginal distribution of spatial frequencies, we see a non-uniform distribution with more higher frequencies than lower ones, increasing the capacity for reconstruction of fine details for super-resolution. We further visualize the average predicted magnitude ("coefficient") of each basis function for an example sequence (Vid4, *calendar*), using a color map in the left subplot of Fig. 6. Magnitudes decrease with increasing frequency, consistent with results from classical Fourier analysis.

### 4.5 COMPUTATIONAL COMPLEXITY

Table 5 lists computational requirements in terms of inference time and VRAM, for various methods. Measurements have been performed on a canonical input patch size of shape $14 \times 80 \times 80$ with upsampling factors of $\times 8$ temporally and $\times 4$ spatially, on an RTX 3090 Ti GPU.

Table 4: Temporal consistency of recent C-STVSR methods in terms of tOF↓ (evaluated on Vid4, $\times 4$ spatial and $\times 2$ temporal super-resolution).

| Bicubic | VideoINR | MoTIF | BF-STVSR | $V^3$ | $V^3$-Large |
|---------|----------|-------|----------|-------|-------------|
| 0.595 | 0.344 | 0.354 | 0.323 | 0.254 | **0.250** |

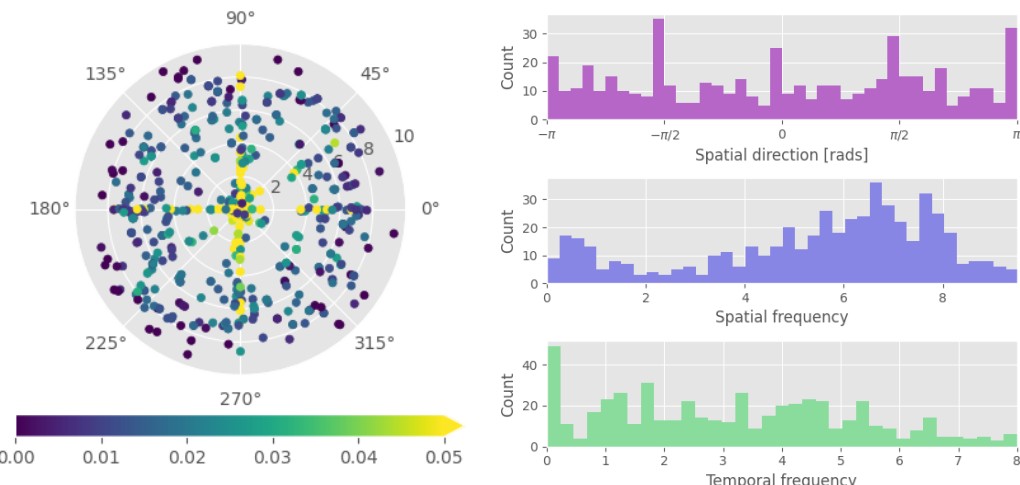

Figure 6: Analysis of globally learned basis functions. *Left*: Polar plot visualizing spatial direction vs. spatial frequency (encoded as radius) of the learned basis functions. The color map visualizes average predicted magnitudes for each component for an example video. *Right*: Marginal distributions of spatial direction, spatial frequency, and temporal frequency of basis functions.

Table 5: Computational complexity of various C-STVSR methods.

|  | VideoINR | MoTIF | BF-STVSR | $V^3$ |
|---|---|---|---|---|
| Inference time | 3.03 s | 1.88 s | 1.90 s | **1.27 s** |
| VRAM | **2.6 GiB** | 8.4 GiB | 10.4 GiB | 6.1 GiB |

## 5 LIMITATIONS AND FUTURE WORK

At very high scaling factors, the outputs of $V^3$ will tend to be overly smooth, a limitation shared by all regression-based SR methods, as a consequence of the discriminative training objective, which favors low distortion over perceptual realism. Generative models (e.g., based on denoising diffusion) would likely deliver results that are more visually pleasing, despite spurious details. In other words, the absence of hallucinated details is a price to pay for low reconstruction error.

The VFF parameterization – a finite 3D Fourier sum – is rather simple and could theoretically present a representational bottleneck in the presence of extensive high-frequency content. So far we did not observe problems for the videos and scaling factors we tested. We suppose that the chosen number of basis functions ($N = 512$), in combination with the local fitting to small $(x, y, t)$-voxels, is sufficient for practical purposes. Note also, it is straightforward to increase $N$ if needed.

Finally, it may be useful to extend $V^3$ to more advanced video degradations beyond spatio-temporal downsampling. E.g., the degradation operator could be used to accomodate sensor noise, motion blur or compression artifacts. Most obviously, VFF would be a natural match for any local, linear degradation – for instance, motion blur is achieved by simply setting $\sigma_{\text{time}} \gg 0$.

## 6 CONCLUSION

We have presented $V^3$, a novel scheme for continuous space-time video super-resolution. Its core component is VFF, a principled, compact video representation in continuous $(x, y, t)$-space based on a Fourier-like 3D frequency decomposition. By combining VFF with a contemporary neural video encoder to predict its free parameters, we construct a clean, unified spatio-temporal super-resolution method that can upsample videos by arbitrary scaling factors in both space and time. $V^3$ exhibits excellent practical performance and outperforms competing methods by $>1$ dB of PSNR, while at the same time offering faster runtime.

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

## A   ADDITIONAL QUALITATIVE RESULTS

Figures 7, 8, 9, 10 and 13 show additional qualitative C-STVSR results for various datasets. Figures 11 and 12 show additional spatial- and temporal-only SR results, respectively.

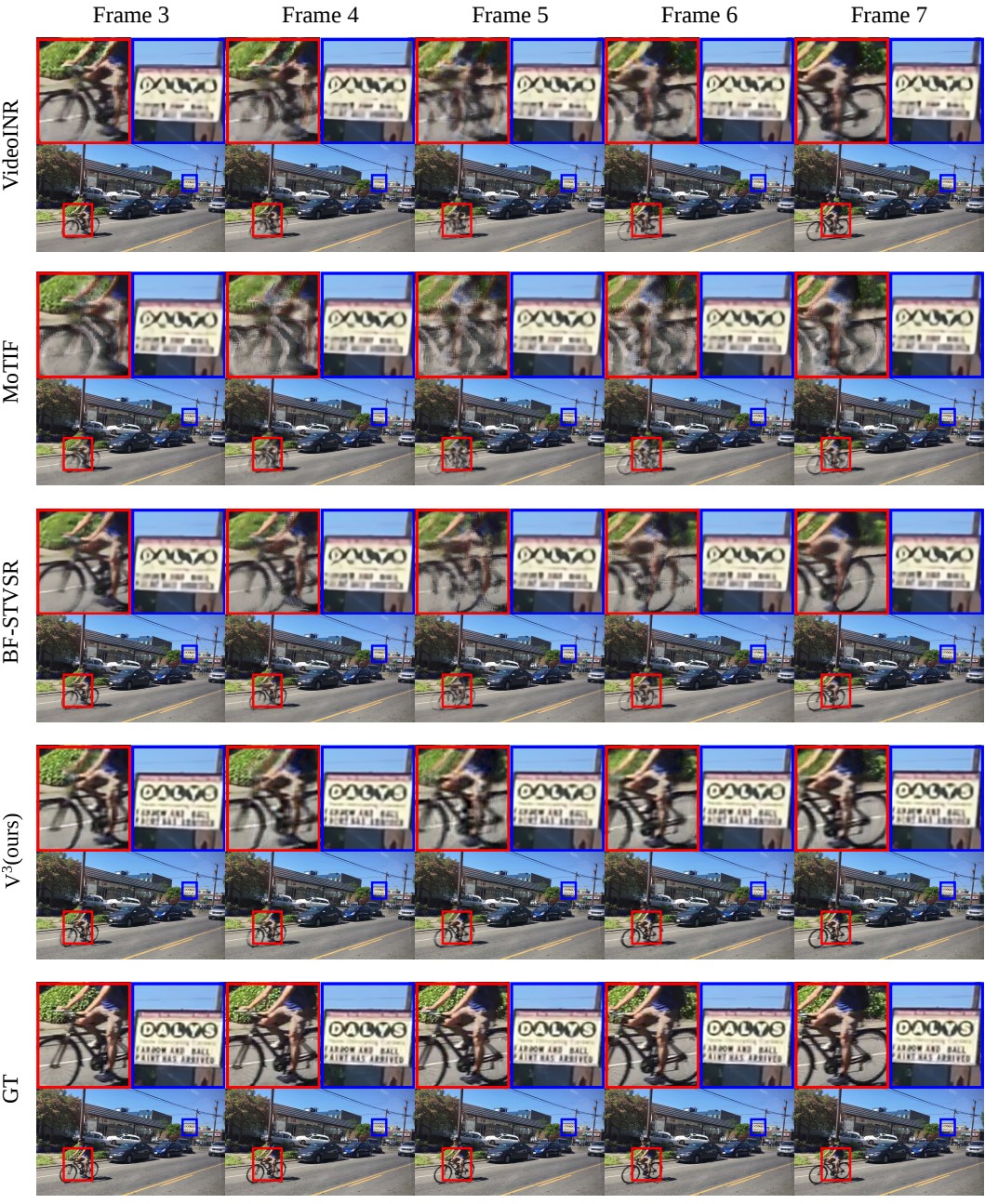

Figure 7: Qualitative comparison of various C-STVSR methods at different time steps on Adobe240. The spatial scaling factor is set to ×4 and the temporal factor to ×8. *Best viewed zoomed in.*

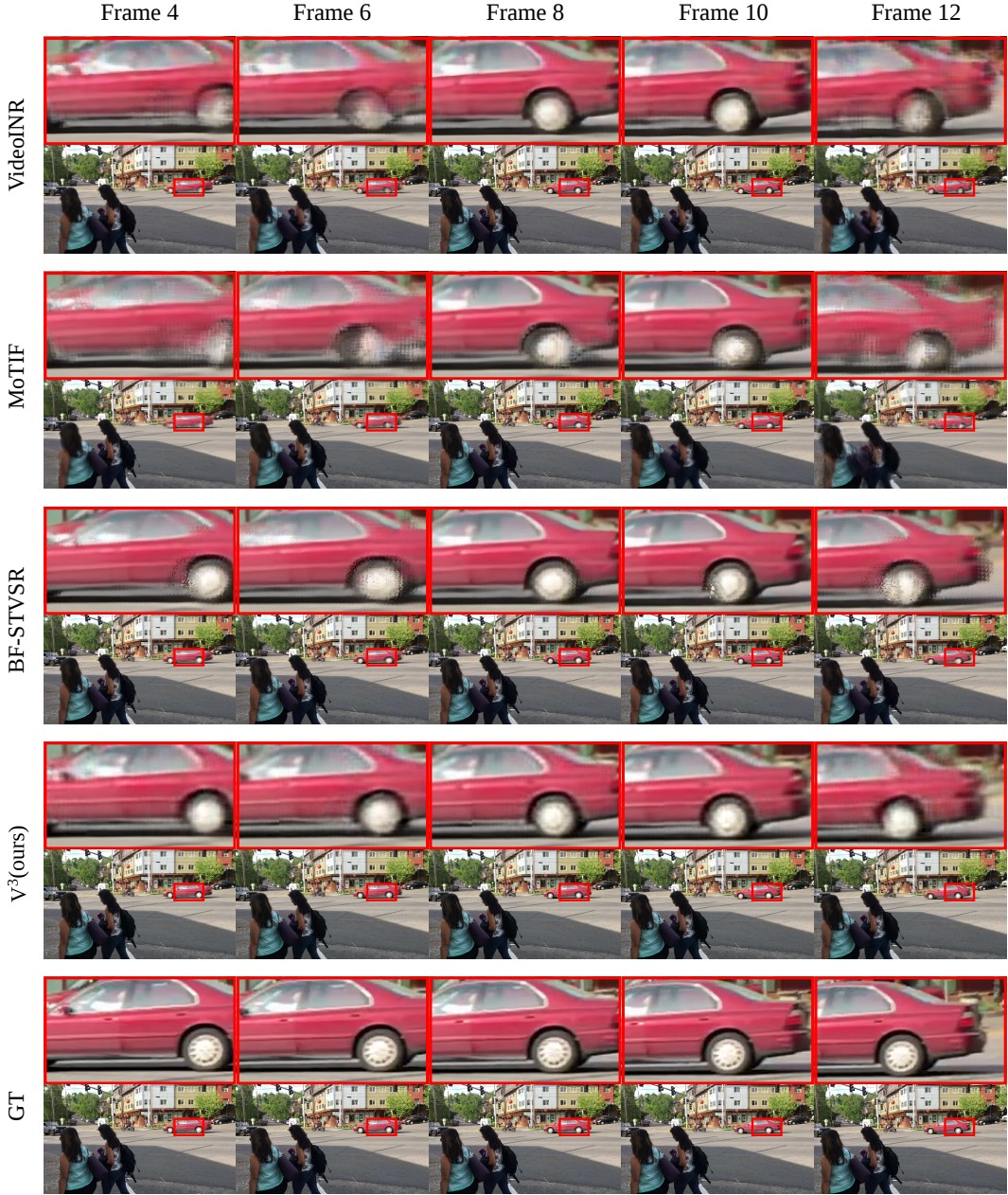

Figure 8: Qualitative comparison of various C-STVSR methods at different time steps on Adobe240. The spatial scaling factor is set to ×4 and the temporal factor to ×8. *Best viewed zoomed in.*

# B ADDITIONAL QUANTITATIVE RESULTS

## B.1 COMPARISON WITH AVSR METHODS

As a complement to Sec. 4.2.1, we also train our method specifically for the AVSR task, enabling direct comparison with prior work (Li et al., 2024; Shang et al., 2024). We denote this variant as $V^{2.5}$, which is architecturally identical to $V^3$-Large[1], but trained on the REDS training set without temporal upsampling and supervised solely on the temporal coordinates of the input frames.

---

[1]With the number of basis functions reduced to $N = 384$.

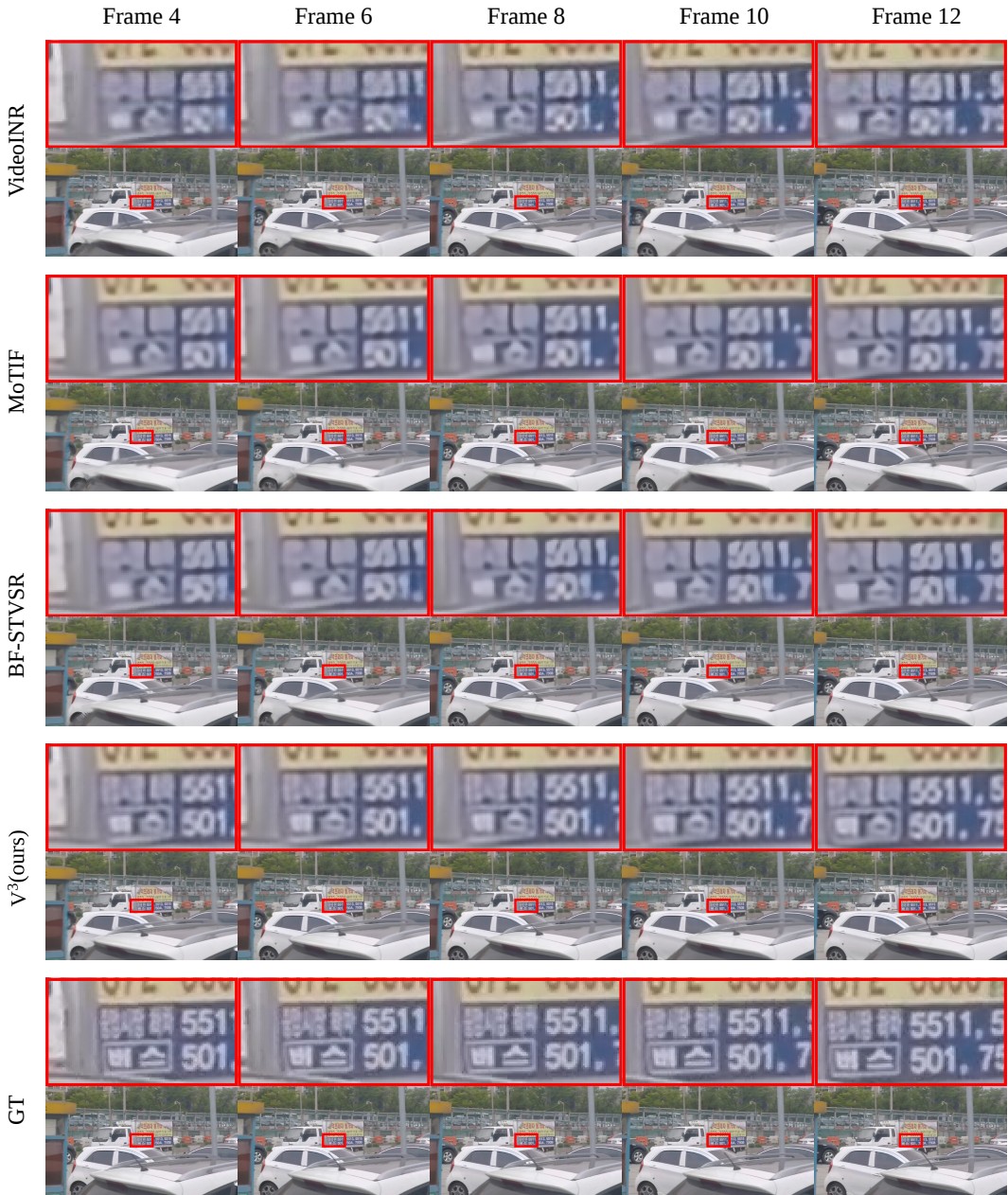

Figure 9: Qualitative comparison of various C-STVSR methods at different time steps on GoPro. The spatial scaling factor is set to ×4 and the temporal factor to ×8. *Best viewed zoomed in.*

Table 6 reports PSNR and SSIM metrics for multiple scaling factors on the REDS validation set. Although not explicitly designed for AVSR – our model remains a more expressive C-STVSR method – $V^{2.5}$ clearly surpasses dedicated AVSR approaches in PSNR. For example, at ×2 scaling it exceeds the next best method by nearly 1 dB. In SSIM, it is only outperformed by ST-AVSR (Shang et al., 2024), which leverages an auxiliary pre-trained VGG16 encoder to extract multi-scale structural and textural features. It is worth noting that ST-AVSR also provides a baseline variant (B-AVSR) without this auxiliary encoder. Since $V^{2.5}$ likewise does not rely on external feature extractors, B-AVSR arguably offers the most direct comparison in terms of training data. Related qualitative samples are provided in Fig. 14.

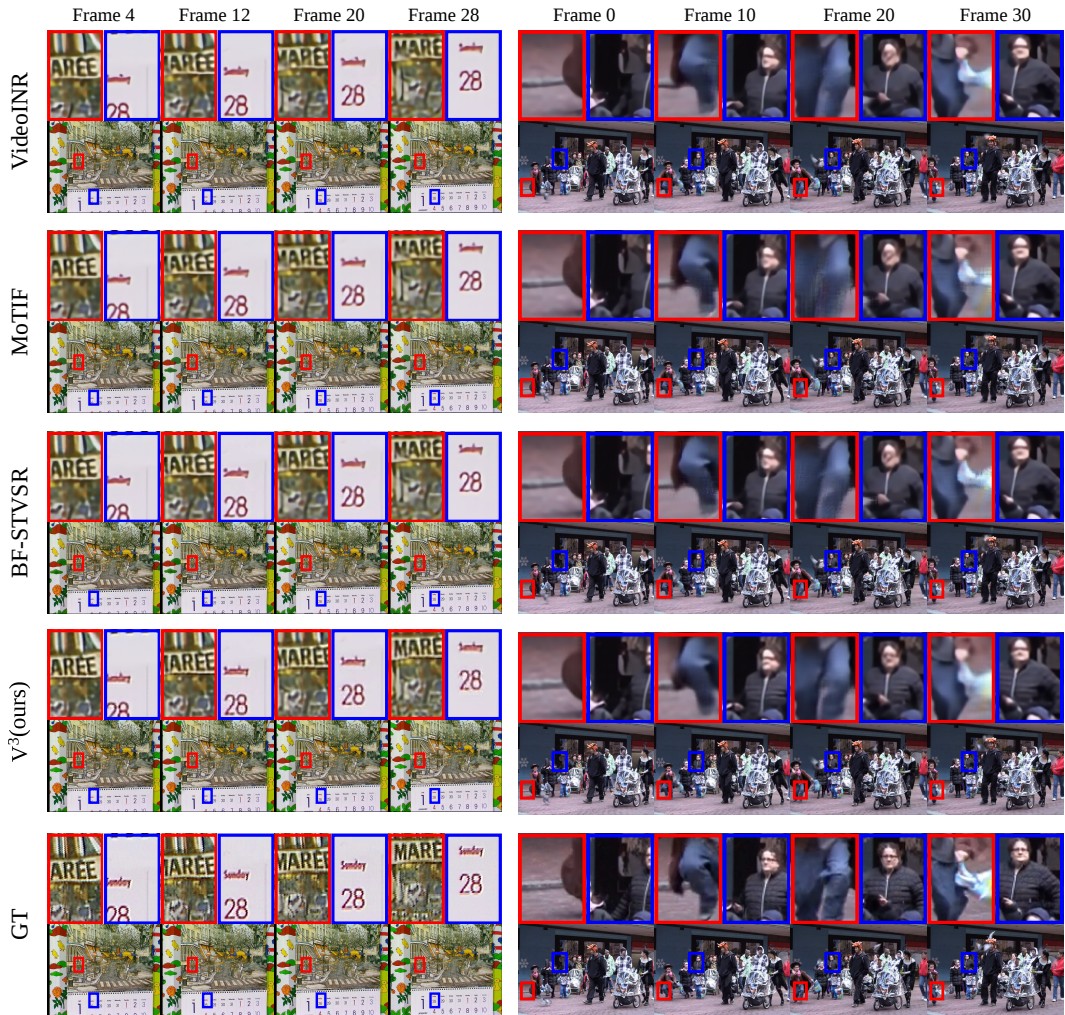

Figure 10: Qualitative comparison of various C-STVSR methods at different time steps on Vid4. The spatial scaling factor is set to ×4 and the temporal factor to ×2. *Best viewed zoomed in.*

## B.2 COMPARISON WITH VSR METHODS

We go one step further and compare $V^{2.5}$ to standard video super-resolution (VSR) methods. In contrast to AVSR, VSR methods do not support arbitrary spatial scaling factors and are specifically trained for a single upsampling rate (usually ×4). In Tab. 7, we compare $V^{2.5}$ with BasicVSR (Chan et al., 2021), TTVSR (Liu et al., 2022), and FTVSR (Qiu et al., 2022). We find that our method achieves the highest PSNR among all compared models and matches the best-performing approach in SSIM, despite being the only method supporting arbitrary-scale upsampling.

## B.3 ABLATION STUDY

In Tab. 8 we analyze several design choices of our method. To isolate the effect of the VFF capacity, we vary the size of the basis while keeping the rest of the model fixed. *I.e.*, we chose different numbers $N$ of basis functions and retrain the mapping from the 256-dimensional encoder output as well as the $N$-dimensional VFF basis.

On both Vid4 and Adobe240, a smaller basis ($N = 256$) leads to a noticeable performance drop, which is further exacerbated when reducing to $N = 128$. On the other hand, increasing the dimension of the basis to $N = 768$ only marginally improves performance. We consider $N = 512$ a

| VideoINR | MoTIF | BF-STVSR | V³(ours) | GT |

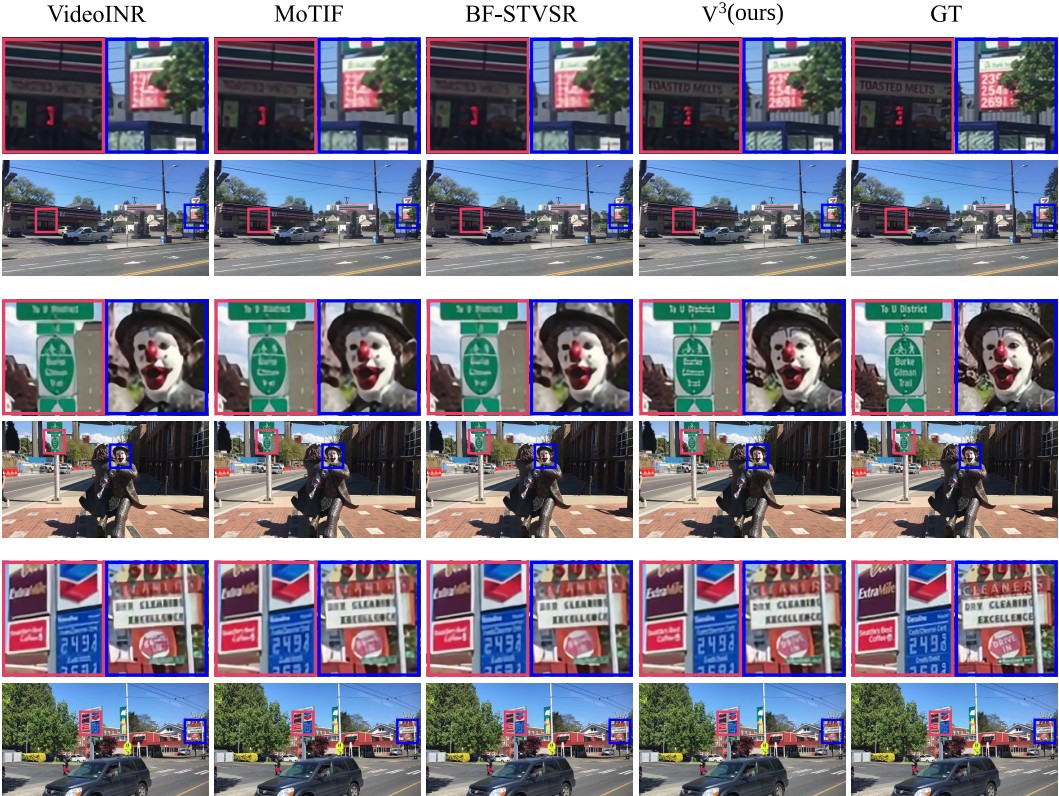

Figure 11: **Spatial Super Resolution** $\times 4$ on Adobe240 dataset.

Table 6: Comparison of methods trained on the AVSR task (PSNR↑ / SSIM↑) on the REDS validation set, with various spatial scaling factors. We re-trained SAVSR on REDS for fair comparison. Also note that ST-AVSR (*) leverages an auxiliary pre-trained VGG-16 model for their structural and textural prior.

| Method | $\times 2$ | $\times 3$ | $\times 4$ | $\times 6$ | $\times 8$ | # Par. |
|---|---|---|---|---|---|---|
| SAVSR | 36.20 / 0.959 | 32.18 / 0.901 | 29.89 / 0.843 | 27.13 / 0.742 | 25.53 / 0.672 | 18.9 M |
| B-AVSR | 35.94 / 0.960 | 31.86 / 0.910 | 29.67 / 0.861 | 26.83 / 0.771 | 25.13 / 0.706 | 14 M |
| ST-AVSR* | 36.91 / 0.969 | 33.41 / **0.937** | 31.03 / **0.897** | 27.89 / **0.812** | 26.04 / **0.746** | 27.9 M |
| V$^{2.5}$ | **37.89 / 0.970** | **33.70** / 0.927 | **31.20** / 0.876 | **28.39** / 0.788 | **26.80** / 0.725 | 20.6 M |

good compromise, noting that a larger basis may in some applications be warranted, and would still be supported by the current encoder capacity.

We further examine the importance of learning the VFF frequencies by pre-setting $\boldsymbol{\omega}_i$ with a canonical 3D plane-wave Fourier basis. Frequency vectors are sampled uniformly from the 3D Fourier frequency space (k-space) and fixed during training. This variant performs significantly worse, which is expected: sampling 512 vectors uniformly in 3D frequency space results in a very sparse grid – only $\sqrt[3]{512} = 8$ unique frequencies per axis – limiting representation power. In contrast, learning the basis functions globally allows to flexibly distribute a constrained number of basis functions so as to maximize their representational power, exploiting structure in the dataset. This is visible in Fig. 6, where learned frequencies concentrate along coordinate axes and distribute non-uniformly over space–time. Notably, learning the base incurs only a negligible overhead: just $3N$ additional trainable scalars and no extra inference cost.

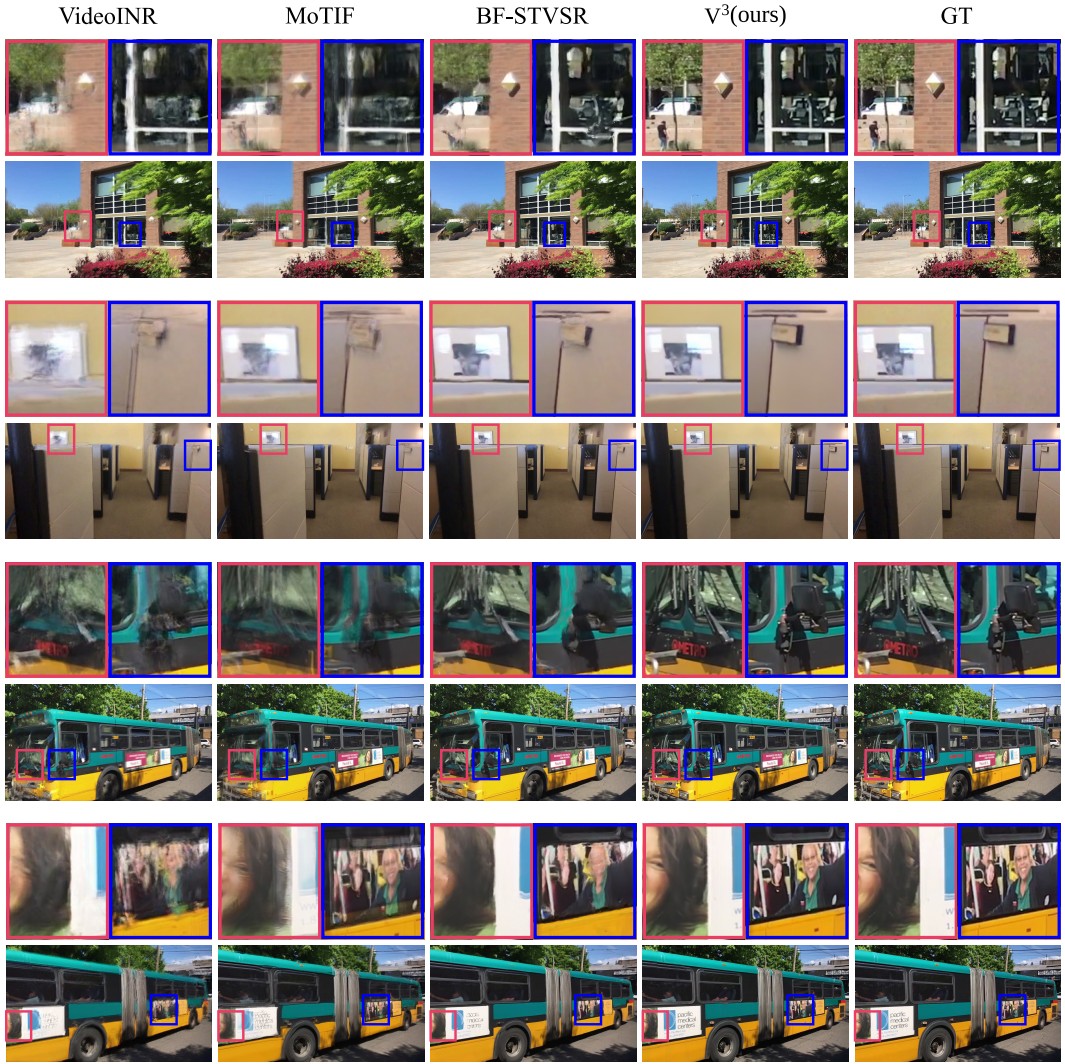

Figure 12: **Video Frame Interpolation** $\times 8$ on Adobe240 dataset.

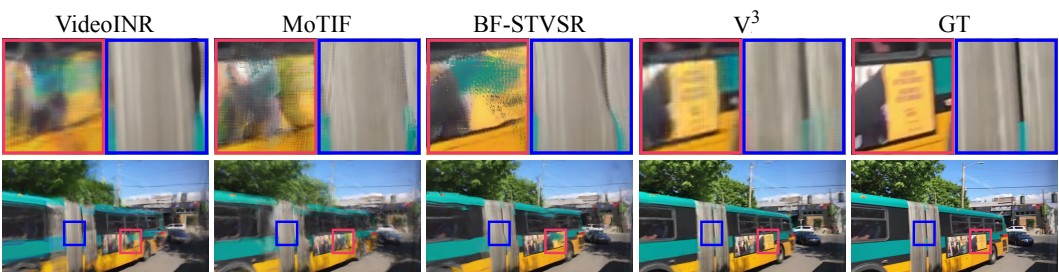

Figure 13: Qualitative comparison of various C-STVSR methods on Adobe240. The spatial scaling factor is set to $\times 4$ and the temporal factor to $\times 8$. We selected an example with large motion blur due to camera motion.

Finally, we ablate the number of training iterations: Reducing to 50% of iterations leads to almost no performance loss (e.g., 26.76 vs. 26.71 dB PSNR on Vid4 $\times 4$). At 25% the drop is more noticeable, but $V^3$ remains the best method by a substantial margin (e.g., $V^3$ at 25%: 31.84 vs. second-best BF-STVSR at 100%: 30.12 dB).

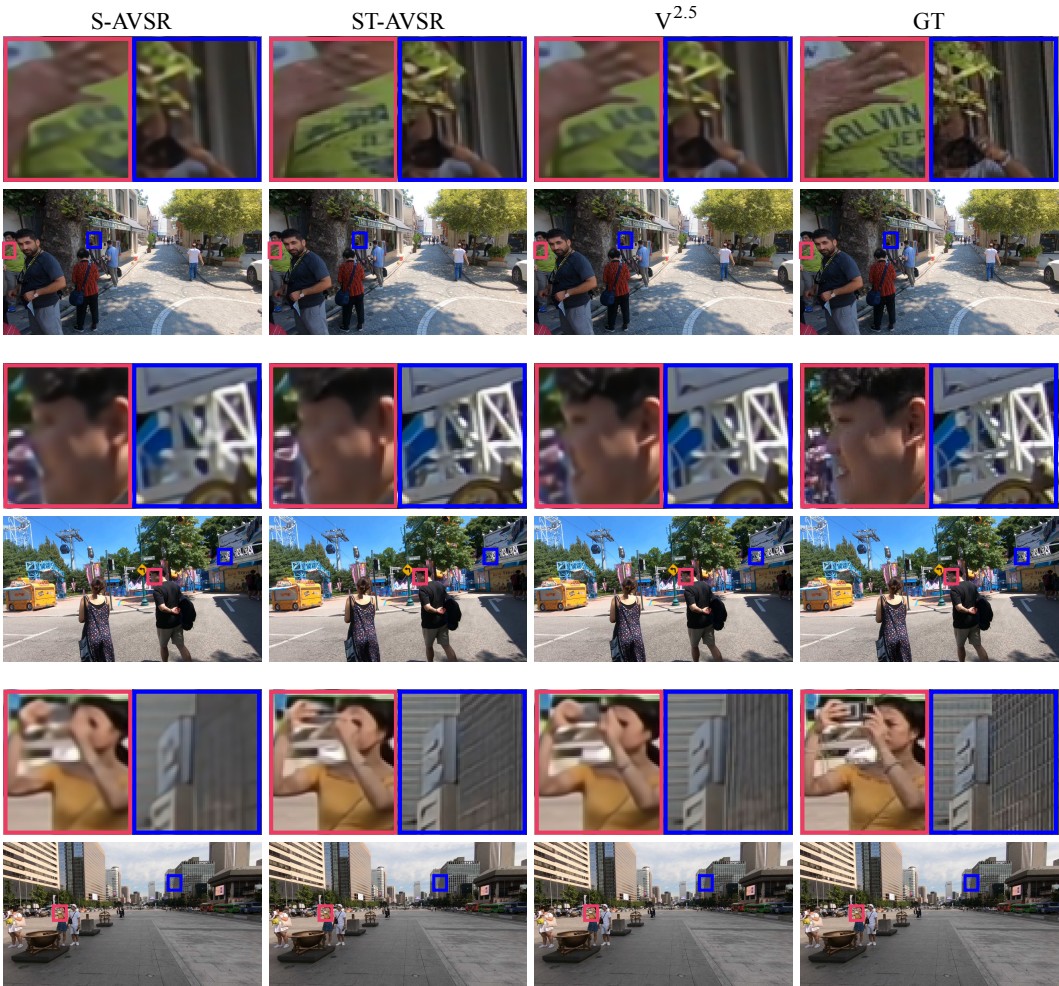

Figure 14: Qualitative comparison of various AVSR methods to our method on ×4 spatial super resolution. Note that no checkpoint for B-AVSR was available at the time of writing.

Table 7: Comparison of our method with various recent video super-resolution (VSR) methods. Note that ours is the only one capable of arbitrary-scale upsampling, while still outperforming others.

|  | PSNR | SSIM |
|---|---|---|
| BasicVSR | 27.24 | 0.825 |
| TTVSR | 27.64 | **0.840** |
| FTVSR | 27.57 | 0.839 |
| $V^{2.5}$ | **27.77** | **0.840** |

## B.4 DEGRADATION STUDY

We analyze two types of image degradation during test time, as shown in Table 9. All experiments are conducted on the Vid4 dataset with a spatial upscaling factor of ×4 and a temporal upscaling factor of ×2. (1) Noise only: We add zero-mean Gaussian noise with a specified standard deviation to images normalized to the range [-1, 1]. The corrupted images are then rescaled to 8-bit [0, 255] and rounded. (2) Noise + compression: We apply the same noise corruption, followed by H.264 video compression of the resulting frames. In both cases, we compare the model's predictions against the non-degraded high-resolution ground-truth images.

Table 8: Ablation experiments on Vid4 and Adobe240 (*average*).

| | Vid4 | | Adobe240 | |
| | PSNR | SSIM | PSNR | SSIM |
|---|---|---|---|---|
| $N = 128$ | 26.19 | 0.797 | 30.55 | 0.884 |
| $N = 256$ | 26.50 | 0.809 | 31.55 | 0.903 |
| $N = 512$ | 26.72 | 0.817 | 32.16 | 0.914 |
| $N = 768$ | 26.74 | 0.818 | 32.29 | 0.917 |
| Fixed $\omega_i$ | 22.85 | 0.651 | 24.80 | 0.738 |
| 50% train iters. | 26.71 | 0.817 | 32.12 | 0.914 |
| 25% train iters. | 26.60 | 0.814 | 31.84 | 0.909 |

Table 9: We ablate how each method's performance degrades under input corruptions on the Vid4 dataset with $\times 4$ spatial and $\times 2$ temporal super resolution. We evaluate two settings: 1) Gaussian noise and 2) Gaussian noise followed by H.264 video compression.

| Method | Degradation: $\mathcal{N}_\sigma$ (Gaussian noise) | | | | |
| | $5 \times 10^{-3}$ | $7.5 \times 10^{-3}$ | $1 \times 10^{-2}$ | $2.5 \times 10^{-2}$ | $5 \times 10^{-2}$ |
|---|---|---|---|---|---|
| VideoINR | 25.55 / 0.766 | 25.50 / 0.763 | 25.45 / 0.759 | 25.04 / 0.729 | 24.35 / 0.677 |
| MoTIF | 25.73 / 0.771 | 25.68 / 0.729 | 25.62 / 0.763 | 25.16 / 0.730 | 24.46 / 0.675 |
| BF-STVSR | 25.82 / 0.774 | 25.77 / 0.771 | 25.71 / 0.767 | 25.28 / 0.735 | 24.53 / 0.679 |
| $V^3$ | **26.68 / 0.814** | **26.50 / 0.806** | **26.28 / 0.796** | **25.58 / 0.757** | **25.18 / 0.721** |

| | Degradation: $\mathcal{C} \circ \mathcal{N}_\sigma$ (H.264 of noised input) | | | | |
| | $5 \times 10^{-3}$ | $7.5 \times 10^{-3}$ | $1 \times 10^{-2}$ | $2.5 \times 10^{-2}$ | $5 \times 10^{-2}$ |
|---|---|---|---|---|---|
| VideoINR | 24.38 / 0.688 | 24.38 / 0.688 | 24.38 / 0.687 | 24.32 / 0.684 | 24.06 / 0.665 |
| MoTIF | 24.54 / 0.690 | 24.53 / 0.690 | 24.53 / 0.690 | 24.47 / 0.686 | 24.21 / 0.666 |
| BF-STVSR | 24.64 / 0.694 | 24.63 / 0.694 | 24.63 / 0.694 | 24.57 / 0.691 | 24.29 / 0.670 |
| $V^3$ | **25.35 / 0.726** | **25.35 / 0.726** | **25.34 / 0.726** | **25.29 / 0.722** | **25.02 / 0.706** |

## C    USE OF LARGE LANGUAGE MODELS

LLMs were used exclusively for text polishing and grammar refinement.

