# OpenReview forum: "Continuous Space-Time Video Super-Resolution with 3D Fourier Fields"
_ICLR.cc/2026/Conference — ICLR 2026 Poster_

### Official Review · Reviewer_hNzY · 2025-10-20

**Soundness:** 3
**Presentation:** 4
**Contribution:** 3
**Rating:** 6
**Confidence:** 4

**Summary:**

This paper introduced INR framework  for arbitrary video super resolution. In particular, the authors utilize 3D Fourier feature embedding to resolve aliasing components during reconstruction. By using 3D representation, they are able to achieve better reconstruction results compared with 2D Fourier embedding or 2D Fourier embedding + B-spline embedding.

**Strengths:**

- 3D video Fourier Field is proposed to reconstruct 3D arbitary super resolution in video. This is easy to handle and simply explained.
- The proposed VFF shows high jump in quantitative comparisons in Tables 1 & 2. It's meaningful gap.
- Temporal consistency looks very coherent in Fig. 5. The core deliver for vsr is usually accurate prediction of temporal coherency. In that terms, V^3 algorithm shows optimal solution for video super resolution.

**Weaknesses:**

- I would like to see computational complexity including memory consumption and inference time as table in main text.
- This is regression model, so when we see the results related with texts in Fig.10, we couldn't guess the text information. Nevertheless, the proposed method is highly impactful in quality perspective.
- no detailed analysis on Fourier coefficients. It will be helpful to show Fourier features aligned with video contents such as DFT spectra of the original contents and estimated coefficients.
- No controlling method for weighting spatial or temporal axis, independently. For example, temperature methods sometimes boiled down the effect of reconstruction. Here in VFF, there is no induced effect by user side. VFF automatically estimated Fourier coefficients along temporal and spatial axis.

**Questions:**

- How does this algorithm behaves for large motion gap such as big fluctuation in optical flow? or abrupt frame changes(key-frame)? In my guess, V^3 depends on 3D Fourier representation which includes time domain resulting in further aliasing components along with temporal axis when they encounter abrupt or big motion changes along with temporal axis

- Isn't there any aliasing results in slomo reconstruction? A task related with slomo is focusing on temporal super resolution, so VFF only provides temporal enhancements which may contradict with spatial super resolution.

---

> ### Author Response · Authors · 2025-11-20
>
> We thank the reviewer for their thoughtful feedback and for acknowledging the suitability of using frequency-based fields for video super resolution, and highlighting the superior temporal consistency induced by it.
>
> ## Computational complexity in main text
>
> We have moved the table from the appendix to the main text (Tab. 5).
>
> ## Properties of per-pixel regression models
>
> Thank you for the remark. Indeed, regression-based approaches with a per-pixel objective are trained to predict an MMSE estimate and can therefore look slightly over-smooth. This comes at the advantage of generally higher fidelity to the ground truth in terms of distortion metrics (e.g. PSNR/SSIM) [1, 2], which is often preferred in super-resolution when results are used for downstream applications (think e.g., facial recognition).
>
> In contrast, generative approaches commit to a single plausible result that may be visually more pleasing, but almost invariably has higher distortions where unrecoverable high-frequency content is hallucinated. We therefore adopt a discriminative objective, like prior C-STVSR methods.
>
> ## Analysis of learned basis functions & coefficients
>
> The V3 basis consists of a set of continuous 3D sinusoidal functions with freely learned frequencies and phase shifts, rather than a classical orthogonal Fourier basis with discrete integer frequencies. Nevertheless, we agree it is interesting to study its distribution in the frequency domain.
>
> We have added Fig. 6 in the revised paper to analyze the spatiotemporal behavior of the learned basis functions. Overall, the **distribution of frequency components is consistent with classical Fourier analysis** of natural images and videos. In particular, we observe a mild concentration of components along the coordinate axes (representing axis-aligned structures), as well as a larger number of components at higher spatial frequencies (representing fine details for super-resolution).
>
> The color map of the left subfigure of Fig. 6 additionally visualizes the **average predicted magnitude of coefficients** for each basis function for an example sequence (Vid4, calendar). The amplitudes decrease systematically with increasing frequency, again consistent with classical Fourier theory, which predicts a decay of spectral energy at higher frequencies.
>
>
> ## Independent control of spatial & temporal modeling
>
> The design goal of VFF is to provide a unified and self-consistent spatio-temporal representation that does not rely on manual weighting between spatial and temporal axes. Unlike factorized approaches, where separate spatial and temporal components naturally introduce tunable trade-offs, our formulation models (x,y,t) jointly within a single Fourier field. The balance between spatial detail and temporal smoothness is therefore learned end-to-end from data rather than exposed as a user-controlled parameter. While axis-specific modulation could be introduced as an extension, we intentionally keep the method free of user-dependent tuning to maintain simplicity and coherence.
>
> ## Behavior under large motion fluctuations
>
> We thank the reviewer for raising this point. A key advantage of our coherent 3D Fourier representation is that it **does not** rely on explicit flow warping, which is known to be brittle under large motion fluctuations or near key-frames. This makes our method more robust in exactly the situations the reviewer describes, and can be seen in, e.g., Figs. 4 & 12: V3 is the only method that recovers high frequency temporal features, while other methods fail to handle abrupt motion boundaries (see e.g., the region of the hand rail).
>
> ## Question about aliasing in temporal SR
>
> We thank the reviewer for the question. In our temporal-only super-resolution experiments, we **do not observe any spatial or temporal aliasing** (see e.g., qualitative samples in Figs. 4 & 12, and Tab. 3 (bottom row)). Moreover, temporal enhancement does not contradict spatial enhancement in our formulation. The unified (x,y,t) representation, together with our multi-frame backbone, aggregates information across many input frames, which in practice benefits both temporal and spatial super-resolution.
>
> ### References
>
> [1] Blau, Yochai, and Tomer Michaeli. "The perception-distortion tradeoff." Proceedings of the IEEE conference on computer vision and pattern recognition. 2018.
>
> [2] Delbracio, Mauricio, and Peyman Milanfar. "Inversion by Direct Iteration: An Alternative to Denoising Diffusion for Image Restoration." Transactions on Machine Learning Research. 2023.

---

> > ### Comment · Reviewer_hNzY · 2025-11-24
> >
> > Thank you to the author for addressing questions and weakness.
> >
> > I will maintain my positive rating.

---

### Official Review · Reviewer_H6t6 · 2025-10-28

**Soundness:** 3
**Presentation:** 3
**Contribution:** 2
**Rating:** 6
**Confidence:** 5

**Summary:**

The paper proposes V3, a novel method for continuous space-time video super-resolution (C-STVSR), centered around a new representation called the Video Fourier Field (VFF). VFF models a video as a continuous 3D trigonometric expansion over joint spatial and temporal coordinates (x,y,t), avoiding the common practice of decoupling space and time via optical flow and explicit frame warping.
Key contributions include:
VFF: A unified, interpretable, continuous video representation using a finite sum of 3D sinusoidal basis functions.
End-to-end framework (V3): A neural encoder predicts VFF coefficients from low-resolution input, enabling arbitrary spatial and temporal upsampling.
Analytical anti-aliasing: Incorporates a closed-form Gaussian point spread function (PSF) for aliasing-free reconstruction at any scale.
State-of-the-art performance: V3 outperforms prior C-STVSR methods by >1.5 dB PSNR across multiple benchmarks while being faster and more memory-efficient.

**Strengths:**

This paper introduces a completely novel approach by modeling video as a joint 3D Fourier field, establishing a new paradigm in the field of video super-resolution.

This paper leverages the classical principle that "translational motion manifests as phase shifts in the frequency domain" and builds a model that does not rely on traditional optical flow algorithms.

The experimental validation is comprehensive: extensive tests on multiple tasks (such as continuous spatiotemporal super-resolution, spatial super-resolution, and video frame interpolation) and various public datasets consistently demonstrate that its performance significantly surpasses existing state-of-the-art methods, with a notable improvement (PSNR gain exceeding 1.5–2 dB).

**Weaknesses:**

The paper employs a globally fixed set of frequencies during training, with only independent modulation of amplitude and phase for each voxel. While this enhances stability, it may limit the model's adaptability to diverse motion spectra (e.g., the differences between high-frequency vibrations and slow translations). The paper does not analyze what specific frequencies are learned or explore how these frequencies correlate with different types of motion. Providing ablation studies or visualizations of the learned frequency distributions would more convincingly demonstrate the expressive power of these basis functions.

There is a lack of comparison and review of classical data-driven methods, such as BasicVSR, TTVSR, and FTVSR.

The degradation model is overly simplified: training assumes ideal bicubic downsampling and clean inter-frame subsampling. However, real-world videos often contain noise, compression artifacts, or motion blur. Although Section 5 mentions this as future work, even a brief experiment (e.g., validating the model's robustness under mild noise or JPEG compression) would more strongly support its claimed practical applicability.

**Questions:**

I'm a bit confused about the method in this paper. The paper proposes modeling a video as a 3D Fourier series, called a Video Fourier Field (VFF). Does this mean that each individual video is represented by its own, independently trained VFF?

This seems different from conventional data-driven video super-resolution (VSR) algorithms. In traditional VSR, once the model is trained, it can theoretically take any input video and produce a high-resolution output.

If, in this paper, every video requires its own dedicated VFF representation, how can this approach be applied in real-world scenarios?

---

> ### Author Response · Authors · 2025-11-20
>
> We thank the reviewer for the positive assessment and insightful comments. We appreciate the recognition that our work introduces a novel 3D Fourier-based video representation and achieves state-of-the-art performance with strong empirical validation.
>
> ## Analysis of learned frequencies
>
> We thank the reviewer for the thoughtful question. Indeed, we chose a global set of basis functions to enhance stability and for its conceptual simplicity.
>
> We agree that further analysis is warranted, and have added Sec. 4.4 and Fig. 6 in the revised paper. The figure provides a statistical overview of converged frequency components of V3. Overall, the learned spatiotemporal frequency distribution is broad and spans a dense range of low-, mid-, and high-frequency variations, demonstrating adaptability to different motion types, from slow translations to fast, high-frequency dynamics.
>
> The figure further shows how the spectra pick up patterns in the data: basis functions concentrate along the coordinate axes (corresponding to horizontal and vertical wave patterns of the rectilinear pixel grid and the prevalence of horizontal and vertical structure in man-made environments). We also observe an over-representation of higher spatial frequencies, corresponding to fine details needed for super-resolution.
>
> ## Comparison with BasicVSR, TTVSR, and FTVSR
>
> We have added comparisons to all VSR methods brought up by the reviewer in Tab. 7 in the Appendix. Despite being the only method to support arbitrary scaling factors, our method achieves the highest PSNR, and matches the best-performing VSR method in SSIM.
>
> ## Simple observation model
>
> For fair comparison, we adopted the same degradation-free data generation protocol as prior C-STVSR (and AVSR/AISR) works. We agree that this is an interesting point, and have added a dedicated degradation study in Tab. 9 in the revision. We evaluate the sensitivity of various methods to 1) Gaussian noise and 2) Gaussian noise followed by H.264 compression. We find that our method is more robust to both types of degradation and still outperforms the best baseline by >0.7dB in PSNR under strong corruption from noise and compression.
>
> ## Question about the method
>
> It is correct that each video is represented by a unique (grid of) VFF -- but, the VFF are **predicted by the backbone at inference time**. There is no video-specific training/optimization at inference time, just a single forward pass as in traditional methods: the encoder predicts a grid of VFF parametrizations, which are then queried at spatiotemporal coordinates to form the upsampled output video. Once trained, V3 can be applied to any input video in pure feed-forward fashion.

---

> > ### Comment · Reviewer_H6t6 · 2025-11-24
> > **repspone**
> >
> > 1. Thank you to the author for the newly added analysis regarding frequency, which helps in understanding the method.
> > 2. The addition of comparisons with classical methods in the field of VSR has enhanced the quality of the paper.
> > 3. The validation in the context of compressed videos has expanded the application scenarios of the proposed method.
> > 4. Thanks for the further explanation.
> >
> > I will maintain my positive rating.

---

### Official Review · Reviewer_MxXA · 2025-10-30

**Soundness:** 3
**Presentation:** 2
**Contribution:** 3
**Rating:** 6
**Confidence:** 4

**Summary:**

This paper introduces a novel formulation for continuous space-time video super-resolution. Unlike previous approaches that decouple video representation into separate spatial and temporal components and rely on frame warping for motion compensation, the proposed method encodes video as a continuous, spatiotemporally coherent 3D Video Fourier Field (VFF). The resulting model, dubbed V³, achieves substantial improvements in both spatial and temporal super-resolution and establishes a new state of the art across multiple benchmarks. However, certain issues regarding the experimental verification remain to be addressed.

**Strengths:**

(1) This work represents video as a continuous, spatiotemporally coherent 3D Video Fourier Field, departing from conventional approaches that decouple video representation into separate spatial and temporal components.

(2) The proposed model, V³, substantially improves both spatial and temporal super-resolution performance, setting a new state-of-the-art on multiple benchmarks.

**Weaknesses:**

(1) The training procedure of V³ lacks clarity. Key implementation details are missing, such as the specific form of the loss function and the value of the hyperparameter σ.

(2) The work would benefit from comprehensive ablation studies. At a minimum, a systematic evaluation of the impact of two key design choices is necessary: the number of basis functions (N) and the value of the hyperparameter σ on final model performance.

(3) The experimental comparisons in Section 4.2.1 have several shortcomings. First, comparisons with Arbitrary-Scale Video Super-Resolution (AVSR) methods are absent, and no visual results from AVSR models are provided. Second, the selection of the two image super-resolution models (LTE and CLIT) in Table 2 is not justified in the text, and the models themselves are not properly cited. Furthermore, as LTE and CLIT are from 2023, including comparisons with more recent methods—such as HIIF [1] would strengthen the evaluation.

[1] 	Yuxuan Jiang, et. al. HIIF: Hierarchical Encoding based Implicit Image Function for Continuous Super-resolution. CVPR 2025.

(4) There are minor typographical errors: a comma is missing in Equation (4), and an unintended line break appears on line 396.

**Questions:**

(1) The training details, specifically the loss function and the hyperparameter σ, need to be clearly stated.

(2) The experimental comparisons should be revised to include benchmarks against AVSR methods.

(3) Ablation studies must be added to systematically evaluate the impact of core model parameters, such as the number of basis functions N and the hyperparameter σ.

---

> ### Author Response · Authors · 2025-11-20
>
> We thank the reviewer for the valuable feedback, for highlighting the fundamental differences of our method to related work, and for acknowledging the substantial improvements over a variety of tasks and datasets.
>
> ## Clarification of training procedure
>
> Thank you for pointing this out. Our method is trained with an L1 reconstruction loss, and we have clarified this in the revised paper (Sec. 3.4).
>
> Moreover, $\sigma$ is not a hyper-parameter but is inversely proportional to the effective sampling rate (i.e., desired scaling factor), governed by the Nyquist limit. Lower scaling factors demand a wider filter kernel (supressing higher frequencies more strongly) for correct anti-aliasing. We have made this clearer in the submission.
>
> ## Ablations
>
> We agree that more ablations are always useful, and have added Tab. 8 to investigate the capacity of the shared frequency bank, on Adobe240 and Vid4 test sets. It shows that N=512 presents a sweet spot, with performance dropping noticeably for 256 and 128 components, but increasing only slightly for N=768. Also, we have added an ablation experiment where we use a fixed canonical 3D plane-wave Fourier basis instead of learnable basis vectors, demonstrating that a basis tuned to the data characteristics achieves better performance with fewer basis functions.
>
> ## Comparison to AVSR methods
>
> We provide comparisons of our method -- re-trained on REDS for fair comparison -- versus dedicated AVSR methods in Sec. B.1 in the appendix. Despite not being specifically designed for AVSR (it is a more expressive C-STVSR method), our method clearly outperforms all others in terms of PSNR (e.g., a nearly 1 dB gap to the second-best competitor for $\times$2 upsampling). In terms of SSIM our method is only outperformed by ST-AVSR, which uses an auxiliary VGG feature extractor for structural guidance. As per the reviewer's remark, we have now added a corresponding qualitative comparison in Fig. 14.
>
> ## Comparison to AISR methods
>
> We initially selected methods representative of two generations of AISR approaches: An earlier CNN-based method (LTE) and a more modern transformer-based one (CLIT). We now explain this reasoning in the text and cite all methods used for comparison. We have also added HIIF, its performance is in line with previous AISR methods and confirms the importance of feature extraction from multiple frames.
>
> ## Minor typographical errors
>
> Thank you for spotting this, we have fixed it in the current revision.

---

> > ### Comment · Reviewer_MxXA · 2025-11-24
> > **Official Comment by Reviewer MxXA**
> >
> > Thank you for addressing my concerns. I am satisfied with the revisions made to the paper and have decided to maintain my positive rating.

---

### Official Review · Reviewer_wVPd · 2025-11-01

**Soundness:** 3
**Presentation:** 3
**Contribution:** 3
**Rating:** 6
**Confidence:** 4

**Summary:**

The paper proposes a continuous space–time video super-resolution (C-STVSR) framework that represents an entire video as a spatio-temporally coherent 3D “Video Fourier Field” (VFF)—a finite sum of sinusoidal basis functions. This avoids decoupling space and time and eliminates brittle frame warping. A neural encoder with a large spatio-temporal receptive field predicts local frequencies, phases, and weights for these basis functions from the low-resolution input, enabling sampling at arbitrary spatial and temporal scales.  The VFF can be queried with a closed-form Gaussian point-spread function, yielding aliasing-free reconstruction by frequency-dependent rescaling rather than expensive filtering, which also generalizes well across scales.  Compared to prior C-STVSR methods and local INR approaches, the unified 3D trigonometric expansion captures long-range temporal dependencies, handles occlusions more robustly, and naturally models translation as phase shifts. Experiments show sharper, more temporally consistent results and about a 2 dB PSNR gain on Adobe240 test set, while being faster.

**Strengths:**

- A continuous-domain video representation that models videos as a single finite trigonometric expansion over (x, y, t) (“Video Fourier Field”), enabling arbitrary spatio-temporal sampling, encoding translation as phase shifts, and supporting analytic, aliasing-free queries.

- The proposed end-to-end C-STVSR framework (V3) whose backbone aggregates over a large spatio-temporal receptive field to predict VFF parameters, leveraging a shared frequency basis for efficiency and stability, and a PSF-aware sampler; the whole system is differentiable and trained end-to-end.

- Strong empirical results: consistent PSNR/SSIM gains across scales on REDS and markedly better video frame interpolation on Adobe240, yielding sharper, more temporally coherent outputs and avoiding warping artifacts typical of optical-flow-based baselines.

**Weaknesses:**

- Since the entire (x,y,t)-space is partitioned into local, axis-aligned regions, does this induce any discontinuities across region boundaries?

- The description of Sec. 3.3 Conditional Fourier Parameterization is unclear. Is the voxel grid for the local basis expansion sample-specific or shared across all samples? How should inputs with different spatial resolutions and aspect ratios be handled? What exactly is the relationship between the voxel grid and the Video Fourier Field? Please provide a more detailed explanation.

- After training, is the common frequency basis identical for all samples? Have you considered presetting a bank of frequency bases as anchors of frequency basis?

- Why is training conducted on 16× GH200? If you used the same compute cost in training stage as prior methods, how would the performance be?

- How does the method handle severe motion blur arising from fast or large motions?

**Questions:**

- Given VFF’s continuous spatio-temporal sampling, can the representation be extended to video frame interpolation, prediction and or video generation, while preserving temporal coherence and anti-aliasing guarantees?
- What motivates introducing a Gaussian PSF with tunable variance?
- Why was JAX chosen over PyTorch/TensorFlow?
- Could you suggest an ablation or variant that reduces training cost without degrading final quality?

---

> ### Author Response · Authors · 2025-11-20
>
> We thank the reviewer for acknowledging our strong empirical results and the benefits and novelty of our proposed frequency-based representation. Your feedback is greatly appreciated, and we incorporated it into our manuscript.
>
> ## Discontinuities across field boundaries
>
> In practice we **do not observe discontinuities** across region boundaries, as is supported by the qualitative and quantitative results. Although the VFF are evaluated locally, the encoder that predicts their parameters has a **large spatiotemporal receptive field** and can therefore predict consistent representations over both space and time.
>
> This design is also standard in prior implicit video and image SR models such as VideoINR, MoTIF, and BF-STVSR, which similarly partition space(-time) but rely on a global backbone to enforce continuity.
>
> ## Description of Sec. 3.3
>
> Thank you for the remark.
> (1) The encoder predicts a sample-specific voxel grid of the same spatiotemporal size and aspect ratio (AR) as the input video. The method therefore natively handles inputs of different sizes and ARs, as shown for instance by the Vid4 results.
> (2) Next, each voxel of the predicted grid informs a local VFF. The $N$ available 3D basis functions are learned globally, but each voxel grid cell has individual local amplitudes and phase shifts for those global basis vectors, resulting in the local expansion.
>
> We have improved Sec. 3.3 to make this clearer.
>
> ## Converged frequency basis
>
> Correct, the common frequency basis, $\omega_i$, is learned globally and is therefore shared by all samples (but with local amplitudes and phase shifts, see previous answer).
>
> To give a better intuition, we have visualised these globally learned sinusoidal basis vectors in Sec. 4.4 in the revision (Fig. 6). Their distribution covers different directions and frequencies (in space-time) quite evenly. As expected, there is a certain concentration of basis vectors on horizontal and vertical wave patterns (the $x$- and $y$-axes of the frequency space), a well-known behaviour due to the rectilinear pixel grid.
>
> Furthermore, we have added a supporting ablation study (Tab. 8, "fixed $\omega_i$") in which we pre-set the representation to a canonical 3D plane-wave Fourier basis. Performance drops significantly as a result, supporting our intuition that sampling only the regular, canonical frequencies yields an overly sparse and constrained representation (only $\sqrt[3]{512}=8$ unique frequencies per axis) and would need a lot more components. In contrast, freely learning the components allows V3 to exploit structure in the data and pick the most suitable components. Note that learning the base has a negligible overhead: just $3N$ additional trainable scalars and no extra inference cost.
>
> ## Hardware
>
> Our model has a larger temporal receptive field and processes multiple frames jointly, which are more efficiently handled with large memory during training. To reduce training time we therefore used 16$\times$ GH200 in parallel. The same performance could be reached with smaller hardware, using gradient checkpointing or accumulation -- but it would not only be slower, but likely consume more total energy.
>
> Deployment of our method does **not** depend on big compute. Inference runs with identical accuracy on a single desktop GPU (RTX 3090 / 4090), like competing  methods.
>
> ## Motion blur
>
> In Fig. 13 in the Appendix, we include a qualitative comparison for a challenging example with extreme motion and blur in the input sequence. Our method mostly preserves the blur present in the inputs and does not exhibit obvious failure modes under these conditions. In contrast, other C-STVSR methods struggle with the substantial inter-frame motion and tend to produce noticeable warping artifacts.
>
> ## Extension to video frame interpolation; temporal consistency
>
> Yes. The advantage of our spatio-temporally coherent formulation is exactly that it can be subsampled in space and time independently. To this end, in Sec. 4.2 we decouple spatial and temporal modeling and show the "edge case" results: AVSR (spatial-only, Sec. 4.2.1) and video frame interpolation (temporal-only, Sec. 4.2.2). Specifically, we show that V3 brings a substantial performance gain in video frame interpolation, in Tab. 3 (bottom), and Figs. 4 & 12.
>
> We also argue that the proposed representation aids temporal coherence/consistency -- to this end, Tab. 4 shows improved temporal consistency metrics of V3, and Fig. 5 gives a qualitative example of V3's smooth temporal modeling.

---

> ### Author Response · Authors · 2025-11-20
>
> ## Variance in Gaussian PSF
>
> The Gaussian PSF introduced in Eq. 4 provides an analytically exact and computationally efficient way of observing a continuous VFF at arbitrary scales.
>
> The standard deviation $\sigma$ of the Gaussian blur **is inversely proportional to the scaling factor** due to the Nyquist limit, instead of being tuned. For instance, observing a signal at smaller scaling factors requires sampling with a larger filter size to prevent aliasing. We have clarified this in Sec. 3.2.
>
> ## Motivation of using JAX
>
> We chose JAX during development for practical reasons, particularly its functional style and simple multi-device parallelization. Our method itself is not tied to JAX and can be equivalently implemented in other frameworks such as PyTorch or TensorFlow.
>
> ## Suggestion of cheaper variant without degrading quality
>
> We have added several ablations (Tab. 8) to investigate this.
>
> **Shorter training schedules**: Using only 50% of the original iterations yields virtually identical performance on Vid4 and is only slightly worse on Adobe240. At 25% of the iterations there is a slightly larger drop, but our method still remains the strongest by a wide margin (e.g., on Vid4 26.76 -> 26.60 vs. second best 25.96; on Adobe240 32.24 -> 31.84 vs. 30.12).
>
> **Smaller shared frequency base**: Reducing the basis to N=256 leads to some drop in performance, but V3 remains the best method by a big margin.
>
> We also note that inference remains lightweight and is already cheaper than competing methods.

---

> > ### Comment · Reviewer_wVPd · 2025-11-28
> > **Official Comment**
> >
> > All my concerns are addressed. Thank you for the rebuttal.

---

### Author Response · Authors · 2025-12-01

We sincerely thank the Area Chair for taking over the handling of our paper, and the reviewers for their valuable feedback, thoughtful questions, and their consistently positive assessment of our work. Their comments highlighted our methodological contributions (*all reviewers*), the novelty of the proposed approach (*MxXA*, *H6t6*), and the comprehensiveness (*H6t6*) and strength (*all reviewers*) of our experimental results. Furthermore, we appreciate that each reviewer indicated that their concerns were addressed during the rebuttal. For a better overview, we briefly summarize the main changes made during the rebuttal (highlighted in blue in the revision) below. For detailed responses to each reviewer, please see our individual replies to the reviews.

* Added an analysis of learned basis functions and predicted amplitudes (Sec. 4.4/Fig. 6).
* Added further comparisons with AVSR, AISR, and VSR methods (Sec. 4.2.1/Tab. 2, Sec. B2/Tab. 7, Fig. 14).
* Added requested ablation experiments and a dedicated degradation study (Sec. B3/Tab. 8, Sec. B4/Tab. 9).
* Added additional qualitative comparisons (Fig. 13).
* Moved the computational complexity table to the main text (Tab. 5).
* Clarified several aspects of the method based on reviewer feedback (Sec. 3.2, Sec. 3.4, Sec. 4.2.1).

We thank the reviewers and the Area Chair again for the constructive discussion and careful consideration. We truly appreciate the time invested in helping us improve this work. We are happy to clarify any remaining points if helpful.

---

### Meta-Review · Area_Chair_rcz7 · 2025-12-25

**Summary:**

This paper proposes a continuous space-time video super-resolution framework that represents a video as a coherent 3D Video Fourier Field (VFF), enabling arbitrary spatio-temporal sampling and aliasing-aware reconstruction via an analytical Gaussian PSF, and reports strong improvements over prior methods across multiple benchmarks. Reviewers’ main concerns focused on (i) clarity of the conditional Fourier parameterization and training details, (ii) adequacy of ablations and comparisons (including AVSR/VSR baselines), (iii) robustness under realistic degradations and motion challenges, and (iv) reporting of computational complexity. The rebuttal and revision addressed these points with added analyses/ablations/comparisons and clarified methodology, and all reviewers indicated they would maintain their positive ratings.

**Reviewer Concerns:**

- Reviewer wVPd: Asked about potential discontinuities across local region boundaries, clarity of the conditional Fourier parameterization (voxel grid/sample-specific vs shared; handling varying resolutions/aspect ratios), the shared/global frequency basis, training compute (16× GH200), and motion blur handling. These were addressed via clarifications and added explanations/ablations, and the reviewer confirmed “All my concerns are addressed”.

- Reviewer MxXA: Flagged missing training details (loss form; σ meaning/value), requested systematic ablations on N and σ, and noted shortcomings in comparisons (missing AVSR; justification/citations for AISR baselines; adding newer methods such as HIIF), plus minor typos. The revision clarified training (e.g., loss), explained σ, added ablations/comparisons (including HIIF/AVSR), and fixed typos; the reviewer stated they are satisfied and will maintain their positive rating.

- Reviewer H6t6: Requested analysis of learned frequencies, broader comparisons to classical VSR baselines (e.g., BasicVSR/TTVSR/FTVSR), and a more realistic degradation study beyond ideal downsampling; also asked whether inference requires per-video optimization. The authors added frequency analysis and additional comparisons/degradation experiments and clarified the feed-forward inference, and the reviewer noted the additions improved the paper and will maintain their positive rating.

- Reviewer hNzY: Requested moving computational complexity to main text, deeper Fourier coefficient analysis, discussed lack of explicit user control over spatial vs temporal trade-offs, and raised questions about large motion gaps/aliasing in temporal SR. The authors responded with added complexity reporting and additional analysis/clarifications, and the reviewer thanked the authors and indicated maintaining the positive assessment. Remaining (minor) outstanding points: While the rebuttal addressed the concrete requested items, the broader limitations typical to regression-based SR (e.g., potential oversmoothing/text legibility trade-offs) and the desire for even more extensive real-world degradation coverage remain as future-work directions (raised explicitly in reviews).

**Reviewer Scores:**

Based on the post-rebuttal discussion, I do not expect score changes; reviewers explicitly stated they would maintain their positive ratings.

- wVPd: 6 → 6 (confirmed concerns addressed)

- MxXA: 6 → 6 (satisfied; maintain positive rating)

- H6t6: 6 → 6 (will maintain positive rating)

- hNzY: 6 → 6 (maintains positive assessment)

---

### Decision · Program_Chairs · 2026-01-26

Accept (Poster)